# Histone lactylation couples cellular metabolism with developmental gene regulatory networks

Fjodor Merkuri[1,2,3], Megan Rothstein[1,2,3] & Marcos Simoes-Costa [1,2,3] ✉

Embryonic cells exhibit diverse metabolic states. Recent studies have demonstrated that metabolic reprogramming drives changes in cell identity by affecting gene expression. However, the connection between cellular metabolism and gene expression remains poorly understood. Here we report that glycolysis-regulated histone lactylation couples the metabolic state of embryonic cells with chromatin organization and gene regulatory network (GRN) activation. We found that lactylation marks genomic regions of glycolytic embryonic tissues, like the neural crest (NC) and pre-somitic mesoderm. Histone lactylation occurs in the *loci* of NC genes as these cells upregulate glycolysis. This process promotes the accessibility of active enhancers and the deployment of the NC GRN. Reducing the deposition of the mark by targeting LDHA/B leads to the downregulation of NC genes and the impairment of cell migration. The deposition of lactyl-CoA on histones at NC enhancers is supported by a mechanism that involves transcription factors SOX9 and YAP/TEAD. These findings define an epigenetic mechanism that integrates cellular metabolism with the GRNs that orchestrate embryonic development.

Cell diversification involves the deployment of tissue-specific GRNs that determine cell identity and function[1,2]. Differential gene expression is thus a key requirement for proper development. However, embryonic cell types also exhibit heterogeneity in non-genetic factors, such as metabolism[3,4]. Embryonic cells often undergo metabolic reprogramming to become suited for specific functions at various developmental stages[5]. For example, the metabolic state of embryonic stem cells (ESCs), which must proliferate and maintain pluripotency, is characterized by an increase in the rate of glycolysis, which is reduced as the cells start to differentiate into specific derivatives[5–7]. These metabolic transitions have important consequences for cellular functions, as inhibition of glycolysis in pluripotent stem cells (PSCs) interferes with their ability to maintain a pluripotent state[7]. Furthermore, in vitro studies of PSC fate decisions demonstrate that metabolic switching is germ-layer-specific and controlled by the dynamics of specific transcription factors[8].

In addition to temporal differences in cellular metabolism, embryos also exhibit spatial heterogeneity of metabolic states. Chick embryos show different patterns of glucose uptake along the anterior-posterior and dorsal-ventral axes, demonstrating that different embryonic tissues have specific metabolic requirements[3,4]. The pre-somitic mesoderm (PSM) and the neural crest are two embryonic cell types for which metabolism has been identified as an important determinant for proper development and gene expression[3,4,9]. In the PSM, an anterior-to-posterior gradient of glycolytic flux influences activation of the Wnt signaling pathway and modulates cell motility. High levels of glycolysis are also important for migration of neural crest cells (NCCs). Prior to the epithelial-to-mesenchymal transition (EMT), these cells exhibit increased glycolytic flux, which is required for expression of genes involved in cell migration. Disruptions to the metabolic states in both cell types result in developmental defects, highlighting the importance of metabolic transitions for proper gene

[1]Department of Molecular Biology and Genetics, Cornell University, Ithaca, NY, USA. [2]Department of Pathology, Boston Children's Hospital, Boston, MA, USA. [3]Department of Systems Biology, Harvard Medical School, Boston, MA, USA. ✉e-mail: marcos@hms.harvard.edu

expression and cell behavior. Yet, the mechanisms that couple cellular metabolism with activation of developmental GRNs remain largely unexplored.

Overproduction of lactate is a well-known consequence of aerobic glycolysis that also occurs in NCCs[4]. In a 2019 study, Zhang et al. identified glycolysis-derived lysine lactylation (Kla) as a new histone mark that is derived from lactate[10]. The authors used macrophage polarization as a model to study the effects of histone lactylation and showed that M1 macrophages (which engage in aerobic glycolysis) have high levels of lactylation. They found that during the M1 to M2 transition, lactylation was specifically deposited around genes associated with M2 phenotypes. Additionally, data from experiments conducted using recombinant chromatin in a cell-free system showed that histone Kla could directly stimulate gene transcription[10]. The regulatory effects of lactylation on gene expression are further supported in the literature by the finding that H3K18La marks both promoters and active enhancers in a tissue-specific manner that resembles H3K27ac[11]. Furthermore, in a 2022 study, Dai et al. use an in vitro system of induced neurogenesis to survey histone lactylation and crotonylation and find that both marks correlate with gene expression at distinct developmental stages[12].

Given the importance of increased glycolytic flux in developmental processes, we investigated the importance of lactylation for the activation of developmental GRNs. Our results show that lactylation is involved in shaping the epigenomic landscape to promote the expression of genes that are essential for NCC development. We observe that lactylation is deposited at active enhancers of important genes in the NCC GRN and that it promotes chromatin accessibility at these regions. We also find that lactylation is tissue-specific by comparing lactylation signatures between NC and PSM cells. Reducing lactylation levels by targeting expression of lactate dehydrogenases led to the downregulation of lactylated genes and was associated with changes in NCC behavior such as impeded migration. Finally, we elucidate a mechanism for NCC-specific lactylation, which revealed a role for transcriptional regulators SOX9 and YAP/TEAD in contributing to the deposition of the mark.

## Results

### The loci of many NC genes are lactylated upon metabolic transition to increased glycolysis

Migration of NCCs depends on their ability to initiate and maintain high glycolytic flux[4]. This state of increased glycolysis is necessary for the expression of key EMT and migration genes in the NCC GRN (Fig. 1a). Since NCCs produce high levels of lactate[4], we tested whether histone lactylation played a role in the epigenetic remodeling required for GRN activation[13]. First, we performed immunofluorescence (IF) staining for pan lysine lactylation (PanKla) in NCC explants. This analysis revealed that the post-translational modification (PTM) is enriched in the nuclei of NCCs, suggesting extensive lactylation of nuclear proteins (Fig. 1b, c). To assess the spatial distribution of lactylation in the embryo, we performed IF staining for PanKla on Hamburger Hamilton Stage 12 (HH12)[14] transverse cranial tissue sections. We observed that cranial NCCs displayed high levels of lactylation (Fig. 1d). Notably, some neural cells (in the neural tube of the developing head) as well as cells of the notochord also display high levels of lactylation (Fig. 1d).

For a quantitative readout of lactylation levels in NCCs as well as to investigate the temporal dynamics of lactylation we performed flow cytometric analysis of dissociated embryonic tissue at three stages of NCC development. PAX7 was used to label neural plate border cells (NPBCs) in HH6 embryos, while TFAP2B was used to mark delaminating (HH9) and migratory (HH12–13) NCCs in embryonic head tissue. We observed that pre-migratory/delaminating NCCs are among the cells with the highest levels of lactylation in the head (Fig. 1e, Supplementary Data 1 and Supplementary Fig. 1). Furthermore, we found that

average lactylation levels increased significantly upon NCC specification (HH6 to HH9) and were highest at HH9 during EMT (Fig. 1f, Supplementary Data 2 and Supplementary Fig. 1). This increase in lactylation levels from HH6 to HH9 coincides with the increase in glycolytic flux that occurs as NCCs begin to express EMT genes[4]. Further analysis revealed that average lactylation levels decrease in NCCs at HH12–13 once they have migrated extensively in the embryonic head (Fig. 1f). However, more than half of TFAP2B+ NCC at HH12–13 (64.47%) have lactylation levels that are comparable to the majority (91.08%) of PAX7+ NPBCs at HH6 (Supplementary Fig. 1M). This suggests that while some NCCs at HH12–13 have reduced lactylation levels, other cells maintain high levels.

The functional consequences of histone Kla were first characterized in the context of M1 macrophage polarization[10]. To investigate the role of this PTM in NCCs, we mapped the genomic profile of histone lactylation with CUT&RUN in cells dissected from HH9 embryos (Fig. 1g and Supplementary Fig. 2A–D). By using an anti-PanKla antibody[10], we identified 10,912 CUT&RUN peaks, which were mapped to their closest annotated gene (Supplementary Data 3). This analysis revealed that lactylation is deposited predominantly in intergenic regions (42.35%) of the genome while also being found in intronic (34.10%), promoter (20.78%), exonic (2.22%), and UTR (0.55%) regions (Fig. 1h). Upon integrating the PanKla CUT&RUN dataset with ATAC-seq data from HH9 NCCs (Supplementary Fig. 2E–H), we found that the majority of lactylated genomic loci are also accessible, as evidenced by their corresponding ATAC-seq signal (Fig. 1i, j). Our ATAC-seq datasets of NCCs identified 38,250 peaks (Supplementary Fig. 2G), 14% (5409) of which are lactylated.

To characterize genes associated with lactylated cis-regulatory elements, we ranked PanKla+ peaks by their average normalized read depth. This analysis showed that several important NCC genes (e.g., SNAI2, ZEB2, SOX10, etc.) are associated with lactylated regions (Fig. 1k, l). We next performed gene ontology analysis using the genes associated with highly-lactylated regions. Consistent with the increase in PanKla just prior to NCC migration (Fig. 1f), we found an enrichment of terms associated with cell adhesion and migration (Fig. 1m and Supplementary Data 4). Taken together, these results demonstrate that lactylation is a dynamic PTM that is deposited in the loci of genes that are part of the neural crest GRN (Fig. 1l). The genomic distribution of lactylation in NCCs suggest a functional role for this mark downstream of the metabolic shifts observed in this stem cell population.

### Histone lactylation marks active enhancers in NCCs

Given the genomic distribution and accessibility of lactylated loci in NCCs, we next asked if lactylation marks active enhancers. We integrated the PanKla CUT&RUN dataset with previously published H3K27ac CUT&RUN data from NCCs[15]. By intersecting peaksets from both datasets, we observed that about half (5515/10,912) of PanKla peaks overlapped with H3K27ac (Fig. 2a). Similar to all PanKla peaks, the PanKla+/H3K27ac+ peaks were also primarily found in intergenic regions of the genome (Fig. 2b) and contained high ATAC-seq signal (Fig. 2c). To interrogate the involvement of lactylated genomic loci in driving tissue-specific gene expression, we examined PanKla levels peaks associated with NCC genes. We intersected our set of lactylated genes with a previously published NCC gene set (Fig. 2d), which was obtained from a likelihood-ratio-test (LRT) comparing gene expression data from NCCs and whole embryo (WE) cells at six developmental stages[13]. We found that lactylated genes (i.e., genes associated with PanKla peaks) are enriched in the NCC gene set when compared to the WE gene set (Fig. 2e), and that lactylated genes from the NCC gene set have higher cumulative PanKla signal when compared to their WE counterparts (Fig. 2f).

To determine if lactylated cis-regulatory regions could act as enhancers, we tested the activity of PanKla+/H3K27ac+ peaks with reporter assays in transgenic embryos. Highly lactylated cis-regulatory

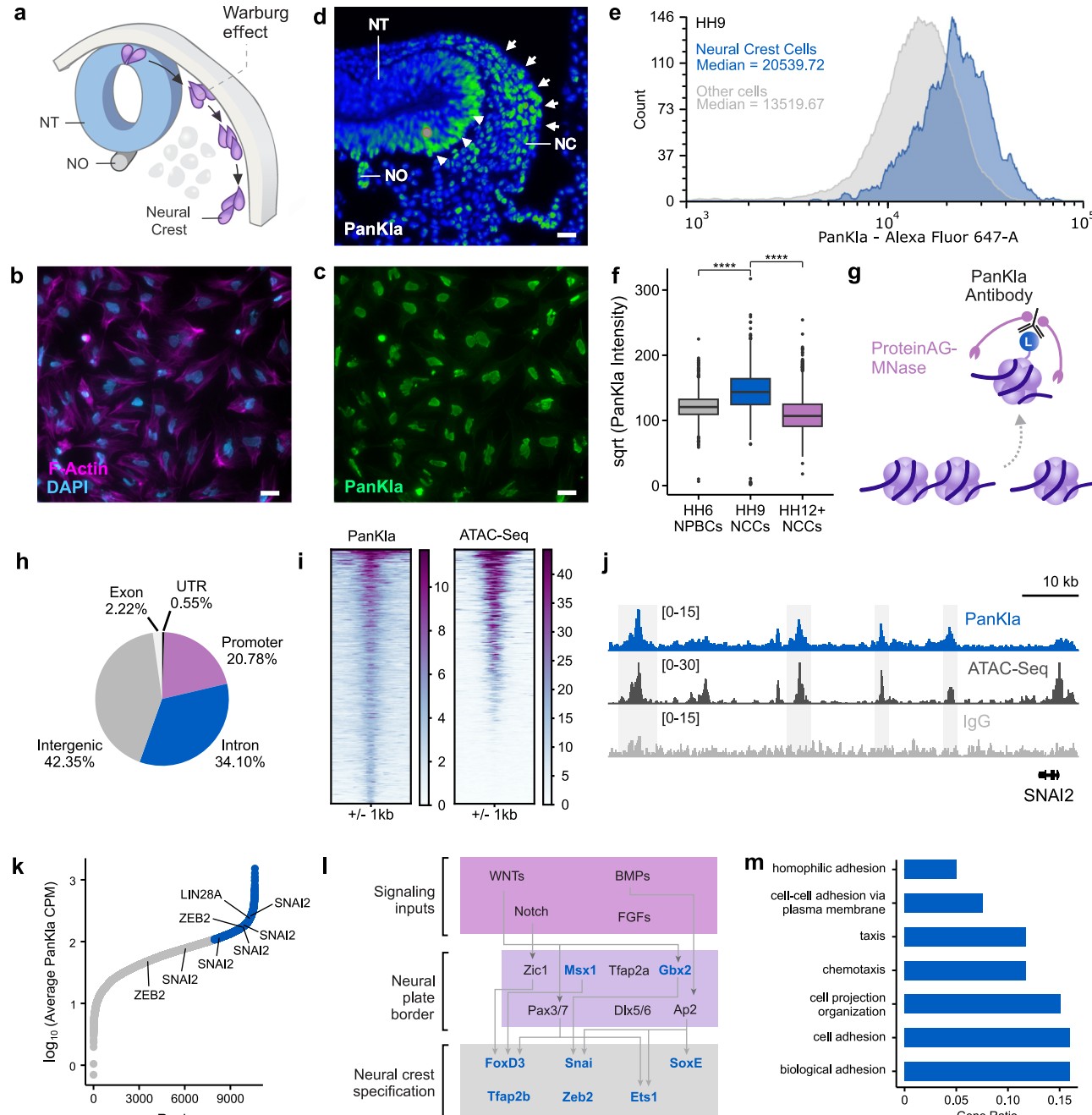

**Fig. 1 | Lactylation exhibits spatiotemporal specificity in NCCs. a** Diagram depicting a cross-section of a chick embryo head with NCC migration path. Increased aerobic glycolysis prior to delamination of NCCs from the neural tube (NT) promotes EMT and migration. The neural tube (NT) and notochord (NO) are also labeled. Reprinted from Bhattacharya et al.[4], with permission from Elsevier. **b, c** IF staining for PanKla of migratory NCCs from explant cultures showing enrichment of lactylation in the nuclei (*n* = 4/4 biological replicates with similar results). Scale bars represent 20 μm. **d** Pseudocolor image of PanKla fluorescence intensity from IF staining on transverse section from HH12 embryonic head displaying the enrichment of lactylation in NCCs (white arrows) and cells in the NT (white arrowheads) (*n* = 2/2 biological replicates with similar results). Scale bar represents 20 μm. **e** Histogram of PanKla fluorescence intensity from the flow cytometric analysis of HH9 embryonic heads. Distribution of lactylation levels in TFAP2B+ NCCs (blue, *n* = 1546) are overlayed on the distribution of lactylation levels in other (TFAP2B-) embryonic head cells (gray, *n* = 15,559). The median for each distribution is shown. **f** Boxplots of square-root-transformed PanKla fluorescence intensity (Alexa647-A) in PAX7+ NPBCs (*n* = 2591) from HH6 embryos and AP2B+ NCCs from HH9 (*n* = 1537) and HH12–13 (*n* = 2471) embryos. ****p value < 2 × 10⁻¹⁶, Kruskal–Wallis test ($\chi^2$ = 2619.7, degrees of freedom (df) = 2), followed by ad hoc pair-wise Wilcoxon rank sum test. The *p* value was

corrected for multiple comparisons using FDR approach. Boxplot center line is median, box limits are upper and lower quartiles, whiskers are the 1.5X interquartile range, and individual points are outliers. **g** Schematic depicting CUT&RUN working principle involving antibody-targeted digestion of chromatin by ProteinAG-MNase fusion protein. **h** Pie chart showing the genomic distribution of PanKla peaks in HH9 NCCs. **i** Tornado plots showing PanKla and ATAC-seq signal at consensus PanKla peakset. **j** Genome browser tracks showing PanKla CUT&RUN and ATAC-seq peaks at the *SNAI2* locus. IgG track included as a control. **k** Scatter plot of consensus PanKla peaks ranked by their average sequencing-depth normalized signal between replicate CUT&RUNs (*n* = 10,612 peaks). Peaks associated with important NCC genes are labeled with the genes they correspond to. Peak with the highest levels of lactylation, upon binning the data, are labeled in blue. **l** Schematic of initial modules of the NCC GRN with genes containing lactylation peaks highlighted in blue. **m** Bar plot displaying a subset of significant (FDR < 0.05) results from the gene ontology enrichment analysis of genes associated with the top third PanKla peaks with highest average signal. Results obtained by using the `enrichGO()` function of the R package clusterProfiler to run a gene ontology over-representation test. RefSeq gene annotation tracks are used to visualize genes in genome browser panels. Non-curated non-coding RNA annotations are not displayed.

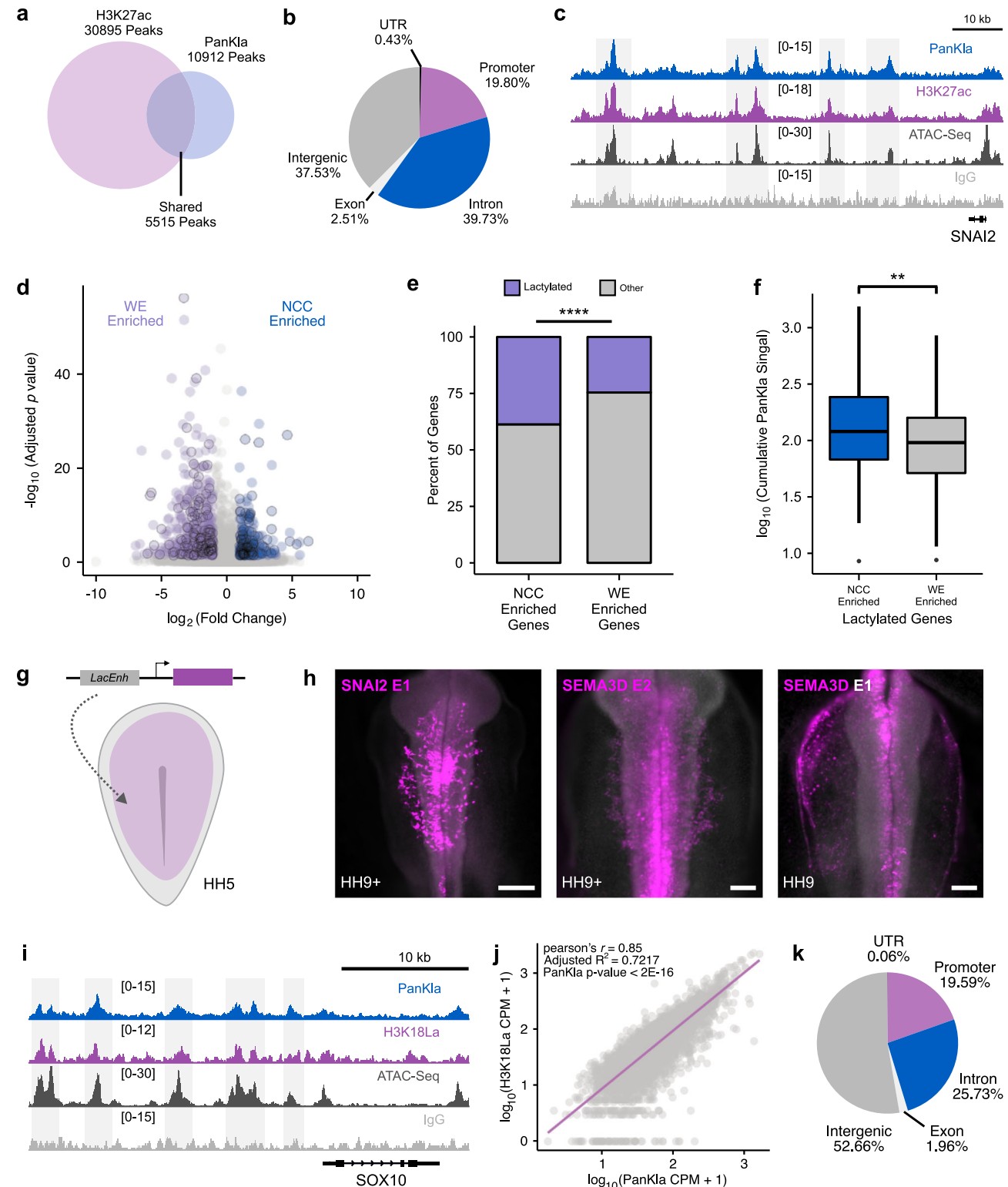

regions from the *SNAI2* and *SEMA3D* genes were cloned in front of a minimal HSV-tk promoter driving GFP[15], and enhancer-reporter constructs were electroporated into the ectoderm of HH4 chicken embryos (Fig. 2g). Reporter expression for the three constructs tested was enriched in NCCs in HH9 embryos (Fig. 2h) demonstrating that PanKla+ cis-regulatory elements can act as NCC-specific enhancers. To analyze the lactylation of NCC enhancers in a systematic manner, we compiled a list of active avian enhancers (in embryonic head and NCC domains) from the literature[15–20] and checked if they were lactylated in

our HH9 PanKla dataset (Supplementary Data 5). We found that 68% of the enhancers (32/47) were lactylated (i.e., overlapped with HH9 PanKla peaks).

These results show that, at least part of, the cis-regulatory elements marked by lactylation in NCCs display enhancer activity. This is consistent with previous reports that lactylation deposited on Lysine 18 of Histone H3 (H3K18La) marks tissue-specific enhancers[11]. Thus, we next performed CUT&RUN for H3K18La in neural folds from HH9 embryos (Supplementary Fig. 2I–L). We found that the levels of this

**Fig. 2 | Lactylation marks active tissue-specific enhancers. a** Venn diagram showing the overlap between consensus H3K27ac and PanKla peaksets from HH9 NCCs. **b** Pie chart showing the genomic distribution of lactylation peaks that overlap with H3K27ac peaks. **c** Genome browser tracks showing PanKla CUT&RUN, H3K27ac CUT&RUN, and ATAC-seq peaks at the *SNAI2* locus. IgG track included as a control. **d** Volcano plot of genes from LRT comparing NCCs to WE cells across six developmental stages. Genes that are significantly enriched in NCCs are labeled in blue whereas genes that are significantly enriched in WE are labeled in purple (FDR < 0.05, $\log_2$FoldChange cutoff set to 1 in both directions). Genes associated with lactylation peaks are outlined in black. **e** Mosaic plot showing the percentage of lactylated genes in the NCC and WE gene sets. Lactylated genes are enriched in the NCC gene set (*n* = 279 total genes) compared to the WE gene set (*n* = 523 total genes). ****$p$ value < 0.0001 ($p$ value = 4.689 × 10⁻⁵), a chi-squared test was used to test for the association of lactylation status and gene set ($\chi^2 = 16.57$, df = 1). **f** Boxplots of the cumulative sequencing-depth normalized PanKla signal of each lactylated gene in the NCC (*n* = 108) and WE (*n* = 129) gene sets. NCC-enriched genes have higher lactylation levels compared to WE-enriched genes. **$p$ value <

0.01 ($p$ value = 0.005882), two-tailed two-sample homoscedastic *t*-test ($t$ = 2.7797, df = 235, 95% CI [0.0423, 0.248]). Boxplot center line is median, box limits are upper and lower quartiles, whiskers are the 1.5X interquartile range, and individual points are outliers. **g** Diagram depicting injection of enhancer reporter construct containing putative lactylated enhancer (LacEnh) cloned upstream of a minimal promoter into HH4 chicken embryo. **h** Embryos expressing GFP reporter driven by sequences underlying lactylation peaks in the *SNAI2* (*n* = 3/3 biological replicates with similar results) and *SEMA3D* genomic loci (*n* = 2/2 biological replicates for each *SEMA3D* enhancer show similar results). SNAI2E1 scale bar represents 200 μm and SEMA3DE2/ SEMA3DE1 scale bar represents 100 μm. **i** Genome browser tracks showing PanKla CUT&RUN, H3K18La CUT&RUN, and ATAC-seq peaks at the *SOX10* locus. IgG track included as a control. **j** Genome-wide correlation of H3K18La and PanKla average sequencing-depth normalized signal at PanKla consensus peakset (*n* = 10,912). Linear regression was used to model the relationship between the two variables. **k** Pie chart showing the genomic distribution of H3K18La peaks. RefSeq gene annotation tracks are used to visualize genes in genome browser panels. Non-curated non-coding RNA annotations are not displayed.

histone-specific lactylation mark were strongly correlated with PanKla levels in NCCs (Fig. 2i, j). Moreover, the genomic distribution of H3K18La peaks was similar to that of PanKla peaks, with a majority located in intergenic regions of the genome (Fig. 2k and Supplementary Data 6). Thus, H3K18La is present in NCCs and is deposited in active regions of the genome. Taken together, these results show that lactylation occurs at tissue-specific enhancers and marks genes that constitute a NCC-specific signature during early development.

## Accessibility of lactylated cis-regulatory elements is spatially and dynamically regulated

We next sought to determine whether lactylated genomic loci are subject to NCC-specific regulation. We performed single cell ATAC-seq (scATAC-seq) and analyzed patterns of genome accessibility in individual cells (Fig. 3a). For this experiment, we dissected the dorsal midline of embryonic heads to obtain a diverse population of cells that includes NCCs as well other neighboring cell populations. The dataset was filtered for high-quality cells, subjected to dimensionality reduction, and clustered to identify distinct population of cells. Analysis of the scATAC-seq dataset identified 7606 high-quality cells (see "Methods" section), which mapped across eight distinct clusters (Fig. 3b and Supplementary Fig. 3A–G). The cellular identity of each cluster was determined by assessing the enrichment of transcription factor binding motifs as well as computed gene scores for cell-type-specific marker genes such as *TFAP2A*, *PAX3*, *OTX2*, and others (Supplementary Fig. 3C, D). This analysis allowed for the identification of various cell populations, including pre-migratory/migratory neural crest, epidermis and cranial placodes, as well as forebrain/midbrain/hindbrain cells. By comparing patterns of accessibility in cluster 7 (C7) and C8 with the other cell types, we were also able to delineate a NCC epigenomic signature. This consisted of peaks that were characterized by transcription factor motifs corresponding to *SOXE*, *TEAD*, and *TFAP2A* (Supplementary Fig. 3E and Supplementary Data 7).

To determine if regulation of lactylated regions is NCC-specific, we projected the accessibility score of PanKla peaks in individual cells of the scATAC-seq UMAP. For this analysis, we focused on a set of cis-regulatory elements (*n* = 5397) that were lactylation-enriched (i.e., PanKla+/H3K27ac−) (Supplementary Fig. 3H), as tissue-specific levels of H3K27ac could confound the results. We found that lactylation-enriched genomic loci were specifically accessible in NCC clusters (C7 and C8) (Fig. 3c). Furthermore, the accessibly of lactylation-enriched peaks was highest in the migratory NCC cluster (C7), consistent with our previous findings that lactylation increases during EMT (Fig. 1f). This led us to examine the accessibility dynamics of lactylated peaks in NCCs. To test whether lactylated *loci* in the genome would become more accessible as NCCs undergo EMT, we used publicly available time-course bulk ATAC-seq data from NCCs isolated at different

developmental stages[13]. This analysis revealed that the cumulative accessibility of HH9 PanKla peaks increases progressively as NCC are specified and begin migrating (HH6–HH10) (Fig. 3d and Supplementary Fig. 3I, J). The same trend in cumulative ATAC-seq signal is also seen at H3K27ac peaks; however, the progressive increase in this context is lower in magnitude when compared to lactylation. This result is consistent with a link between lactylation and the activation of genes and GRN circuits that are involved in EMT and cell migration.

Next, we used pharmacological manipulation of metabolic state to test the importance of lactylation, downstream of increased glycolytic flux, for changes in chromatin organization during NCC development. To this end, we cultured NCC explants in media supplemented with DMSO or (R)-GNE-140 and performed bulk ATAC-seq after 12 h of treatment (Fig. 3e and Supplementary Fig. 4A–H). The drug (R)-GNE-140 is a potent dual inhibitor of the LDHA and LDHB enzymes responsible for synthesizing lactate from pyruvate. To confirm that the explant system did recapitulate the global genome accessibility patterns of NCCs in vivo, we first compared ATAC-seq data from control (DMSO) explants to NCCs extracted from wild-type embryos. The results revealed a high correlation between ATAC-seq profiles of explants and primary NCCs (Supplementary Fig. 4I, J). Comparing the datasets from control and treated explants, at control (baseline) peaks, led to the identification of differentially accessible regions that were either enriched (1187) or depleted (1699) upon (R)-GNE-140 treatment (Fig. 3f, g and Supplementary Data 8). This shows that inhibition of lactate production led to changes in chromatin accessibility of NCC ATAC peaks with certain regions gaining or losing accessibility. As expected, we found an enrichment of lactylated peaks among the ATAC-seq peaks that were significantly depleted in the (R)-GNE-140 condition (Fig. 3g). Furthermore, the cumulative accessibility of PanKla peaks was lower in the (R)-GNE-140 treatment compared to vehicle (Fig. 3h). Taken together, these results show that lactylated genomic *loci* are specifically accessible in NCCs and exhibit dynamic regulation throughout NCC development. We also find that restricting the production of lactylation precursors results in a reduction of accessibility at lactylated genomic loci.

## Lactylation is deposited in a cell-type-specific manner

Our results in NCCs show that lactylation is associated with components of a specific GRN, and support a role for this mark in promoting the accessibility of tissue-specific enhancers. To test if this PTM is deposited in a cell-type-specific manner, we examined lactylation patterns using CUT&RUN in the PSM, another highly glycolytic cell population in vertebrate embryos (Fig. 4a and Supplementary Fig. 5A, B). High glycolytic flux in PSM cells is important for cellular motility and for the activation of the GRN that controls somitogenesis[3,9]. Consistent with the results obtained in NCCs (see "Results" section for

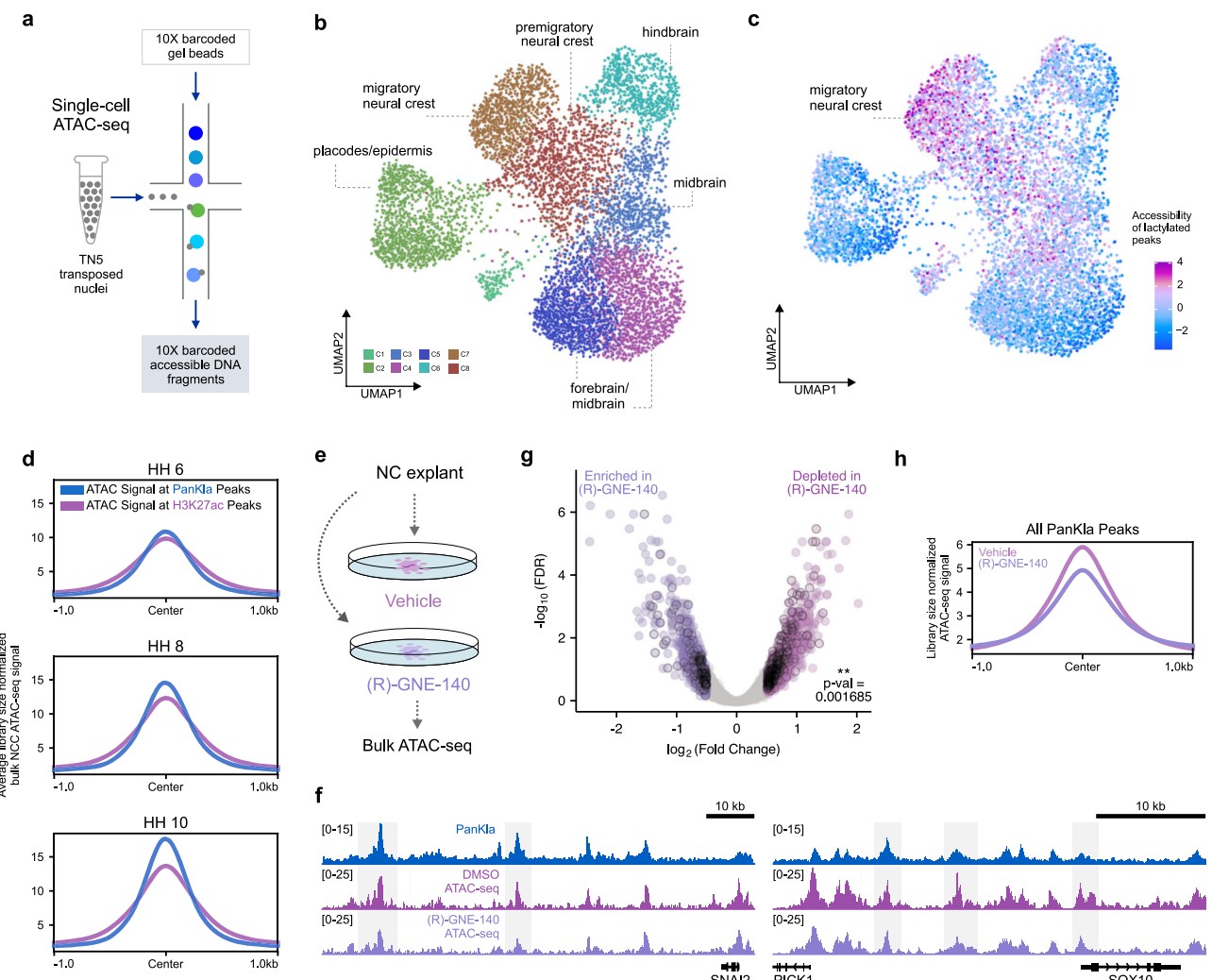

**Fig. 3 | Lactylation promotes accessibility of genomic regions in NCCs.**
**a** Schematic depicting scATAC-seq workflow. **b** UMAP projection of scATAC-seq profiles of cells from the dorsal midline of HH9 embryos. Dots represent individual cells while colors indicate cluster identity (labeled). **c** UMAP projection colored by the degree of accessibility of lactylation-enriched peaks (relative to H3K27ac) from the PanKla CUT&RUN. Lactylated genomic loci are specifically accessible in NCC clusters (C7 and C8). **d** Profile plots showing the average (between two biological replicates) cumulative ATAC-seq signal at consensus peaks from the PanKla (blue) and H3K27ac (magenta) CUT&RUNs. ATAC-seq data are from samples that consist of sorted NCCs at three developmental stages (HH6, HH8, and HH10). The accessibility of lactylated genomic loci increases as NCCs begin to migrate. **e** Schematic depicting treatment of NCC explants from HH9 embryos in defined culture conditions with DMSO or 40 uM (R)-GNE-140. ATAC-seq was performed on explants 12 h after treatment to assess the effects of (R)-GNE-140 treatment on chromatin accessibility. **f** Genome browser tracks showing PanKla CUT&RUN, DMSO and (R)-GNE-140 ATAC-seq peaks at the *SNAI2* and *SOX10* loci. **g** Volcano plot showing differentially accessible control peaks between DMSO and (R)-GNE-140 treatments. Differential accessibility analysis was performed using the DBA_EDGER method within the R package DiffBind. Peaks that are significantly depleted in (R)-GNE-140 are labeled in magenta, whereas peaks that are significantly enriched in (R)-GNE-140 are labeled in purple (log$_2$Fold Change cutoff set to 0.5 in both directions). Lactylated peaks are outlined in black. A chi-squared test was used to test for the association of lactylation status and enrichment/depletion in (R)-GNE-140 ($\chi^2 = 9.8649$, df = 1, *p* value = 0.001685). Of the 1699 ATAC-seq peaks that were depleted in (R)-GNE-140, 269 were lactylated whereas only 138 peaks were lactylated among the 1187 peaks that were enriched in (R)-GNE-140. **h** Profile plots showing the average cumulative ATAC-seq signal of DMSO (magenta) and (R)-GNE-140 (purple) samples at consensus peaks from the PanKla CUT&RUN. RefSeq gene annotation tracks are used to visualize genes in genome browser panels. Non-curated non-coding RNA annotations are not displayed.

Fig. 1), PanKla peaks were located in the *loci* of PSM GRN genes like *FGF8* and *TBXT* (Supplementary Fig. 5G). Comparison between the lactylation signatures of NCCs and PSM identified 3275 lactylated peaks that were specifically enriched in the PSM and 3030 peaks that were enriched in NCCs (Fig. 4b and Supplementary Fig. 5C). These differentially enriched PanKla peaks were associated with cell-type-specific genes (Fig. 4b and Supplementary Data 9). For genes that are expressed in both NCCs and PSM cells (e.g. *SNAI2*[21]), we observed the presence of both shared and tissue-specific peaks (Fig. 4c), with the latter being present in smaller numbers. Next, we performed functional term enrichment analysis on the genes that were associated with differentially enriched peaks in NCCs or PSM cells. This analysis

revealed that genes associated PSM lactylation peaks had an enrichment of terms such as somite development, segmentation, cellular response to BMP, and mesoderm development (Fig. 4d). Conversely, genes found close to NCC lactylation peaks were related to chemotaxis, locomotion, and cellular projection organization (Supplementary Data 10). Interestingly, both NCC and PSM lactylation gene sets were enriched for terms related to cell adhesion (Fig. 4d), indicating that this glycolysis-lactylation axis is important for EMT and cell motility.

To investigate the transcription factor motifs that are associated with lactylation peaks in NCCs and PSM cells we ran chromVAR[22] to determine the variability in lactylation (between NCC and PSM

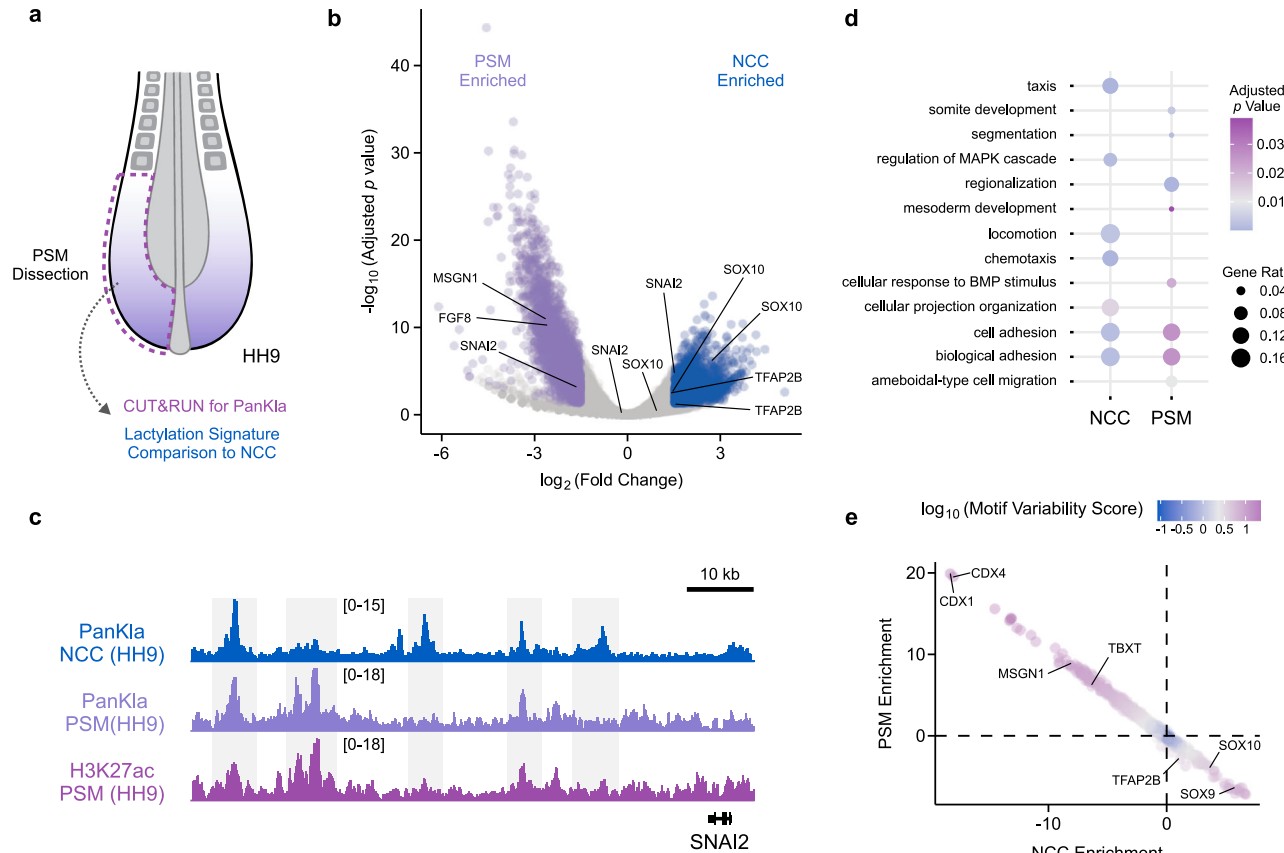

**Fig. 4 | Deposition of lactylation is cell-type specific. a** Diagram depicting dissection of posterior embryonic tissue containing PSM cells. CUT&RUN for PanKla was performed on PSM tissue and the lactylation signature was compared to that of NCCs. **b** Volcano plot of differential lactylation peaks between NC and PSM cells. Genes associated with lactylation peaks are labeled on the plot. Significant (FDR < 0.05) lactylation peaks enriched in NCCs are labeled in blue whereas lactylation peaks enriched in PSM cells are labeled in purple (log$_2$Fold Change cutoff set to 1.5 in both directions, DiffBind concentration score >3). Differential accessibility analysis was performed using the DBA_EDGER method within the R package DiffBind. **c** Genome browser tracks showing NCC and PSM PanKla CUT&RUNs as well as PSM H3K27ac CUT&RUN (Supplementary Fig. 5D–F) at the *SNAI2* locus. **d** Dot plot displaying a subset of significant (FDR < 0.05) results from the gene ontology enrichment analysis of genes associated with NCC or PSM enriched peaks. Results obtained by using the `enrichGO()` function of the R package clusterProfiler to run a gene ontology over-representation test. **e** Scatterplot of chromVAR results showing the differential enrichment of transcription factor binding motifs in NC and PSM cells. Axes represent the average deviation *z*-score of lactylation peaks containing specific motifs in both NCC and PSM samples. Color indicates the variability score of a specific motif in the differential analysis peakset. RefSeq gene annotation tracks are used to visualize genes in genome browser panels. Non-curated non-coding RNA annotations are not displayed.

samples) of sequences that contain known transcription factor motifs. We then plotted the average deviation *z*-score of lactylation peaks containing specific motifs in both PSM and NCCs (Fig. 4e). This analysis revealed that NCC and PSM peaks had enrichment of lineage-specific transcription factor motifs (Supplementary Data 11). For example, motifs for transcription factors such as *CDX* factors, *MSGN1*, and *TBXT* were enriched in the PSM lactylated peaks whereas motifs for factors like *SOX9*, *SOX10*, and *TFAP2B* were enriched in the NCC dataset. Taken together, these results show that lactylation is differentially deposited in distinct cell types, at loci that are regulated by lineage-specific transcription factors. Functional terms associated with lactylated genes in NCCs or PSM cells also indicate that lactylation is deposited in a manner that supports cell-type-specific functions and behaviors (Fig. 4d).

## Glycolysis-derived lactylation is required for activation of the NCC GRN

Glycolysis and endogenous production of lactate are key determinants of histone Kla levels[10]. To gain insight on the importance of histone Kla for cell behavior and gene expression, we knocked down lactate dehydrogenase A and B (*LDHA/B*) using translation-blocking morpholinos (MOs) targeted to the mRNA of each enzyme. To validate the knock-down efficiency of the MOs, we designed a test construct containing the MO-targeted exons fused to an RFP coding sequence. The test construct for each gene was then co-electroporated with control or LDHA/B MOs on each side of a single embryo (Supplementary Fig. 6A). Both LDHA and LDHB test constructs exhibited a drastic decrease in RFP expression when co-electroporated with the LDHA/B MO mix (Supplementary Fig. 6B, C). To confirm that *LDHA/B* knock-down reduced lactylation levels, we performed flow cytometry after staining embryonic head tissue from embryos transfected with FITC-Control and LDHA/B MOs for PanKla (Supplementary Fig. 6D–I). This analysis revealed a reduction of PanKla levels in high-PanKla FITC-positive cells of LDHA/B MO-transfected embryonic tissue compared to control (Supplementary Fig. 6J). Furthermore, cells transfected with LDHA/B MOs also displayed an increase in the proportion of FITC-positive cells with low PanKla levels (Supplementary Fig. 6J).

Having validated the effects of the LDHA/B knock-down on PanKla levels, we measured the effects of PanKla reduction on NCC behavior. We used bilateral electroporation to transfect HH4 embryos with control MO (left side) and LDHA/B MOs (right side) and allowed them to develop until HH9+ (Fig. 5a). We then performed IF staining for TFAP2B to mark cranial NCCs and quantified the migration area on each side of the embryo (Fig. 5b, c). We found that NCC migration was

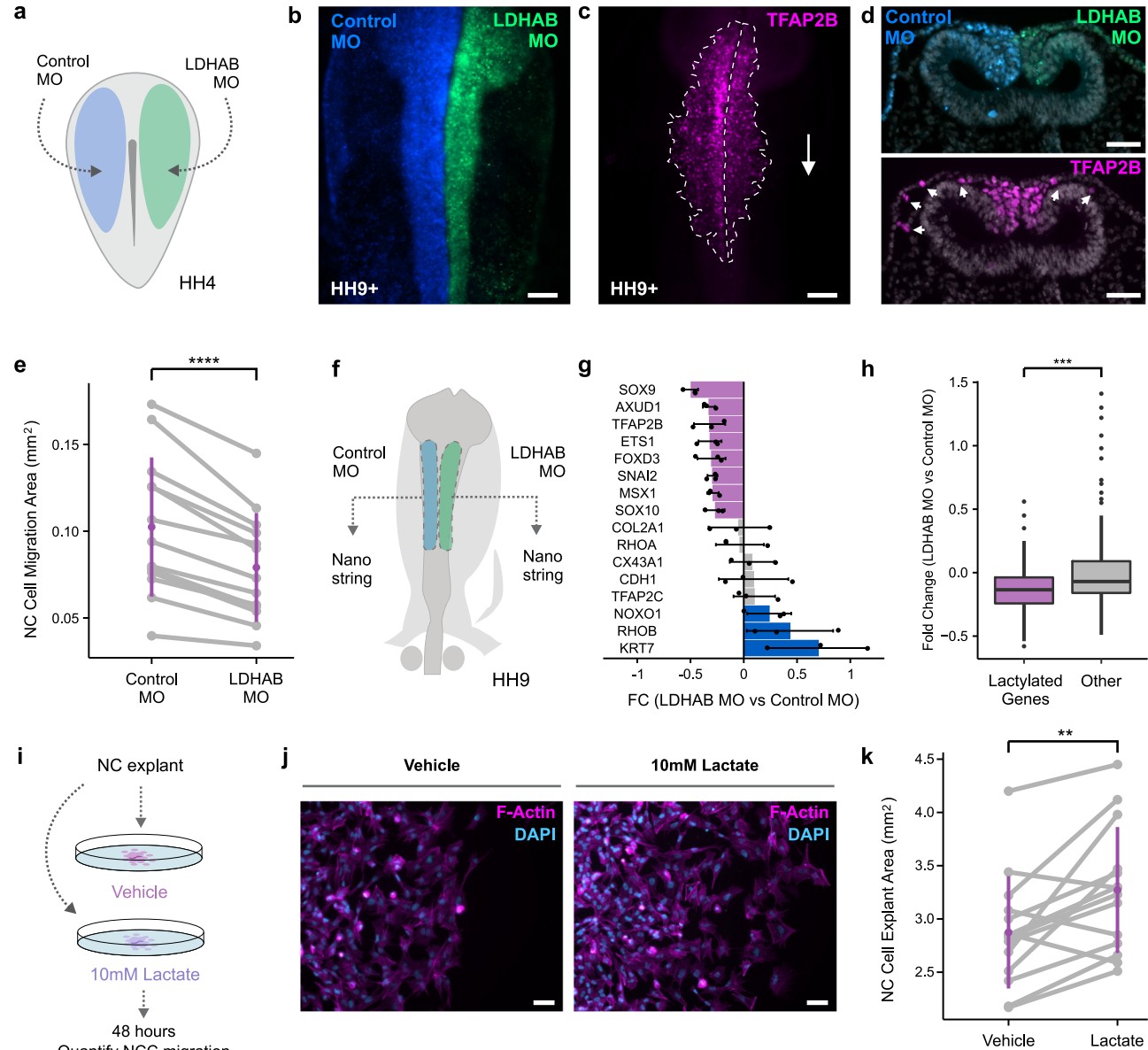

**Fig. 5 | Reducing lactylation through LDHA/B loss-of-function impedes NCC migration. a** Diagram depicting the bilateral injection of control and LDHA/B MOs in HH4 embryos. **b** Representative image of HH9+ embryo transfected with Control and LDHA/B MOs. Scale bar represents 100 μm. **c** IF staining for TFAP2B in embryo from (**b**) showing reduced NCC migration on the LDHA/B-transfected side. Scale bar represents 100 μm. **d** Transverse section of HH9+ embryo transfected with Control and LDHA/B MOs. IF staining for TFAP2B shows reduced NCC migration on the LDHA/B-transfected side compared to Control MO side. Scale bar represents 50 μm. **e** Paired stripchart showing the quantification of NCC migration area from whole mount TFAP2B-stained HH9-HH10 embryos (*n* = 14 biological replicates). Purple bars represent standard deviation. ****p value < 0.0001 (*p* value = 1.004 × 10⁻⁵), two-tailed paired homoscedastic *t*-test (*t* = 7.2574, df = 12, 95% CI [0.0163, 0.0303]). **f** Diagram of HH9 embryo depicting experimental strategy for Nanostring experiment. Control and LDHA/B MO transfected neural folds were collected from the same embryo. Nanostring was performed on single paired neural folds. **g** Bar plot showing fold change (FC) of mRNA levels in LDHA/B MO vs. Control MO samples for important genes in the NCC GRN. Error bars are standard

deviation of three biological replicates (*n* = 3). **h** Boxplots showing the fold change (LDHA/B vs. Control MO) of lactylated (*n* = 76) and non-lactylated, or "other" (*n* = 123) genes in the Nanostring probeset. Lactylated genes are, on average, down-regulated more than other genes in LDHA/B MO neural folds. ***p value < 0.001 (*p* value = 0.0001383), two-tailed independent heteroscedastic *t*-test (*t* = −3.8885, df = 194.76, 95% CI [−0.2255, −0.0737]). Boxplot center line is median, box limits are upper and lower quartiles, whiskers are the 1.5X interquartile range, and individual points are outliers. **i** Diagram depicting experimental strategy for assessing the effects of lactate treatment on NCC migration. Neural fold explants for control and lactate treatment were collected in a paired fashion from the same embryo. **j** Staining of NCC explants treated with vehicle (1X PBS) or 10 mM sodium lactate with Phalloidin and DAPI. Scale bar represents 50 μm. **k** Paired stripchart showing the quantification of NCC explant area (*n* = 14 biological replicates). Purple bars represent standard deviation. **p value < 0.01 (*p* value = 0.005), two-tailed paired homoscedastic *t*-test (*t* = 3.4, df = 13, 95% CI [0.1417, 0.6526]). Graphics in (**a**) and (**f**) were reprinted from Bhattacharya et al.[4], with permission from Elsevier.

significantly reduced in the LDHA/B-transfected side of the embryo compared to the control side (Fig. 5d, e), with no changes in cell death or proliferation (Supplementary Fig. 7A–D). These results demonstrate that endogenous lactate production and glycolytic flux, both important precursors of histone Kla, are required for NCC migration.

To assess the effects of LDHA/B knock-down on gene expression, we performed Nanostring analysis of MO-transfected single neural folds dissected from the same embryo (Fig. 5f). For this experiment we used a probeset designed against genes involved in NCC, neural, and placodal development[23,24]. Upon analyzing the

Nanostring data we found that the expression of several key lacty-lated genes in the NCC GRN was reduced in the LDHA/B MO condition when compared to control (Fig. 5g and Supplementary Data 12). Furthermore, the average fold change of lactylated genes in the probeset was significantly lower than that of non-lactylated genes (Fig. 5h). This shows that LDHA/B knock-down specifically affects the expression of lactylated genes in the NCC GRN. Finally, to test if exogenous lactate supplementation had an opposite effect to LDHA/B knock-down, we cultured neural fold explants in control or lactate-supplemented media for 48 h (Fig. 5i). We then quantified explant area as a proxy for NCC migration and found that lactate treatment enhanced the migratory capacity of NCCs (Fig. 5j, k and Supplementary Fig. 7E, F). Taken together, these results demonstrate that lactylation and its chemical precursors in the cell are required for NCC migration and proper activation of the NCC GRN.

### SOX9 and YAP/TEAD are involved in the cell-type-specific deposition of lactylation marks

Previous studies using chromatin reconstitution assays showed that lactylation is deposited in cis-regulatory elements in a p300-mediated manner[10]. However, how the mark in deposited at cell-type-specific enhancers is still unknown. This prompted us to investigate the mechanism underlying the deposition of lactyl-CoA on histones in the genome of NCCs. We used HOMER to search for transcription factor binding motifs that are enriched among the lactylation peaks from our CUT&RUN dataset. Consistent with our comparisons between NCCs and PSM (see "Results" section for Fig. 4), this analysis revealed strong enrichment of the SOX motif (Fig. 6a and Supplementary Data 13). We also found a significant enrichment of the TEAD motif, which is in agreement with the previous report that YAP/TEAD acts downstream of glycolysis in NCCs[4]. The enrichment of the SOX and TEAD motifs prompted us to analyze the occupancy of associated factors in the context of PanKla in NCCs. We performed CUT&RUN for SOX9 (the SOX factor with one of the highest expression in NCCs) in neural folds from HH9 embryos (Supplementary Fig. 8A–D) and used publicly available CUT&RUN data for active YAP1 in NCCs[4]. Comparing these datasets with our CUT&RUN for PanKla confirmed that lactylated peaks overlap with both YAP1 and SOX9 peaks (Fig. 6b). To test whether the occupancy of YAP1 and SOX9 could predict lactylation levels, we employed a multiple regression model where YAP1 and SOX9 were set as explanatory variables and PanKla was set as the response variable. This model showed that YAP1 and SOX9 were both significant predictors of PanKla levels (Fig. 6c). The model adjusted $R^2$ value shows that the relationship between PanKla and YAP1/SOX9 can explain most of the variation (72%) in PanKla signal.

Next, we asked whether SOX9 is required for maintaining the accessibility of lactylated genomic *loci*. To answer this question, we performed ATAC-seq on HH9 neural folds from embryos transfected with a SOX9 MO (Fig. 6d and Supplementary Fig. 8E–M)[25]. Analyzing this dataset revealed that *SOX9* knock-down led to widespread changes in chromatin accessibility (Fig. 6e) and that ATAC-seq peaks that are depleted in the SOX9 knock-down condition display higher levels of lactylation (Fig. 6f and Supplementary Data 14). Indeed, we find an enrichment of PanKla peaks among the peaks that loose accessibility upon SOX9 MO treatment (Fig. 6g, h). The same trend is observed for SOX9 peaks but not ATAC-seq peaks at promoter regions (Fig. 6h). Next, we tested if we could increase the global levels of lactylation by manipulating the activity of transcriptional regulators in NCCs. Given our motif enrichment analysis of lactylated peaks (Fig. 6a), and functional analysis (Fig. 6e–h), we focused on SOX9 and YAP/TEAD. We performed a gain-of-function (GOF) experiment where HH4 embryos were transfected with a *SOX9* over-expression construct[25] as well as a *TEA1-VPR* construct[4] (Fig. 6i). The embryos were collected at HH6 (prior to the transition to high glycolysis and before the onset of *SOX9* expression in NCC) and the region of the embryo containing the

presumptive NCCs was dissected and subjected to CUT&RUN for PanKla (Supplementary Fig. 9). Analysis of CUT&RUN data revealed that the average normalized read depth for PanKla was higher in SOX9/TEA1 expressing embryos compared to RFP control (Fig. 6j, k). To analyze the data we employed two independent normalization strategies that account for compositional differences between libraries (i.e., Trimmed Mean of *M*-values (TMM)[26] and spike-in normalization to carryover E. coli DNA[27,28]) (Supplementary Fig. 9G–L). Both normalization approaches consistently showed the same trend of increased average PanKla signal in SOX9/TEA1 expressing embryos compared to control (Supplementary Fig. 9). This increased trend is not observed when considering HH6 PanKla peaks from control (RFP samples). These results further confirm that genomic loci containing SOX and TEAD motifs have a propensity to become lactylated in NCC and implicate SOX9 and YAP/TEAD in contributing to the cell-type-specific deposition of lactyl-CoA in NCCs.

## Discussion

Metabolic regulation has emerged as an essential player in cellular identity and behavior during embryonic development. Growing evidence supports robust coupling between the bioenergetic and transcriptional states of developing cells. The present study identified a mechanism centered around histone lactylation that integrates the metabolic state of embryonic cells with the deployment of cell-type-specific GRNs. We observed that this PTM is dynamically deposited in the genome of NCCs as they transition into a state of enhanced glycolysis. This process, among others, is required for epigenomic remodeling and proper expression of developmental genes. Thus, histone lactylation has a similar function to acetylation, which has long been observed to enhance transcription of regions where it is deposited[29,30]. Given the chemical similarities between lactylation and acetylation[10,31], it is likely that the former also weakens the interaction between histones and DNA[32,33], and increases the efficiency with which RNAPII moves through chromatin in vitro[34]. Indeed, histone acylations (e.g., acetylation, crotonylation, and lactylation) have been shown to directly increase the rate of transcription in cell-free assays[10,35]. Our findings indicate that deposition of lactylation in the genome of NCCs is one of the mechanisms downstream of increased glycolytic flux that promotes (1) accessibility of NCC enhancers, (2) expression of NCC genes and (3) cellular behaviors like EMT and migration.

Our results are consistent with recent reports that investigate the functional implications of lactylation. In a 2020 study by Li et al., this PTM was shown to play a role in pluripotency acquisition[36]. Following reprogramming by *GLIS1*, histone lactylation and acetylation levels were shown to increase at the loci of "second wave" and pluripotency genes, thereby facilitating somatic cell reprogramming[36]. The tissue-specific deposition of H3K18La at enhancer elements further supports the regulatory effects of lactylation on gene expression[11]. In a 2022 study, Dai et al. use an in vitro system of induced neurogenesis to survey histone lactylation/crotonylation and investigate their relationship with stage-specific gene expression, finding evidence in support of a functional role for both marks[12]. Notably, the authors demonstrate that global levels of lactylation and crotonylation are not a consequence of transcription by showing that the levels of the PTMs are not affected by RNAPII inhibition. In a 2015 study, Sabari et al. show that histone crotonylation can directly stimulate transcription in a cell-free assay and that increasing the intracellular levels of crotonyl-CoA enhances gene transcription[35]. The results of this study support a model where the concentration of different acyl-CoA species (such as acetyl-CoA and crotonyl-CoA) determine the differential acylation state of chromatin, thereby facilitating transcription to varying degrees. Our results show that a similar mechanism underlies the regulatory role of lactylation on gene transcription, especially since the levels of this mark are subject to change that depends largely on the glycolytic state of the cell.

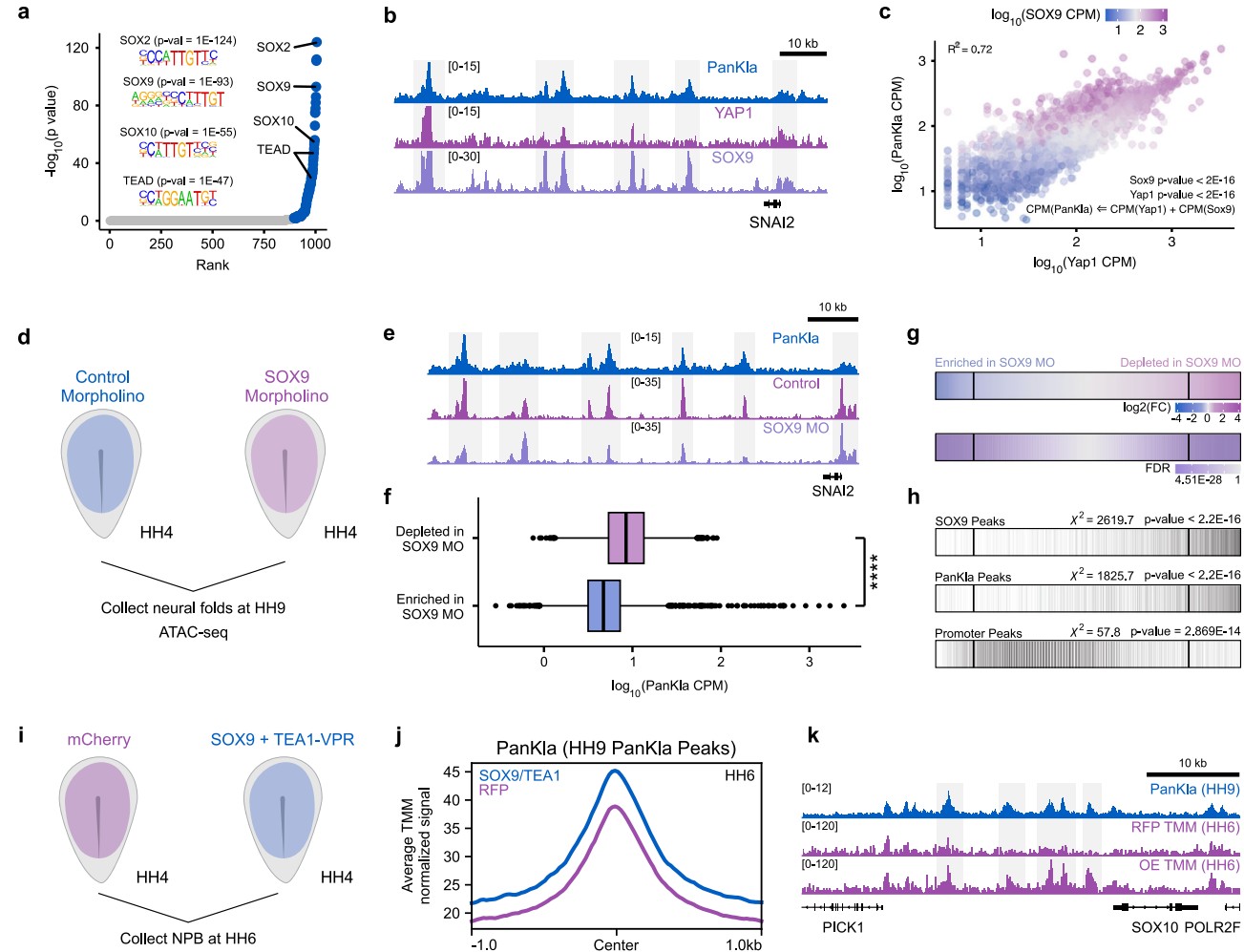

**Fig. 6 | SOX9 and YAP/TEAD contribute to the deposition of lactylation at NCC-specific loci. a** Scatter plot of transcription factor binding motifs enriched among PanKla peaks ranked by *p* value. Motifs with an adjusted *q*-value (Benjamini) less than 0.05 are labeled in blue. A significant enrichment of SOX and TEAD motifs was observed. HOMER was used to perform enriched motif discovery using exact peak sizes. **b** Genome browser tracks showing PanKla, YAP1, and SOX9 CUT&RUNs at the *SNAI2* locus. **c** Scatter plot of sequencing depth normalized YAP1 and PanKla signal at PanKla peaks. Color indicates the level of SOX9 signal. The relationship between YAP1, SOX9, PanKla was modeled using multiple linear regression with YAP1 and SOX9 occupancy as explanatory variables and PanKla as a response variable (*n* = 10,448). **d** Diagram depicting experimental strategy for Control and SOX9 MO injections in HH4 embryos. Neural folds for each condition were then collected at HH9 and subjected to the ATAC-seq workflow. **e** Genome browser tracks showing PanKla CUT&RUN, Control MO and SOX9 MO ATAC-seq peaks at the *SNAI2* locus. **f** Boxplots of sequencing depth normalized lactylation signal at ATAC-seq peaks that were significantly (FDR < 0.05) enriched (*n* = 10,775) or depleted (*n* = 7270) in the SOX9 MO condition compared to Control MO (log₂FC cutoff set to 0.5 in both directions, *n* = 2 biological replicates). ATAC-seq peaks that are depleted in SOX9 MO transfected NCCs have significantly higher lactylation levels. ****$p$ value < 2 × 10⁻¹⁶, two-tailed Wilcoxon rank sum test (W = 57767759). Boxplot center line is median, box limits are upper and lower quartiles, whiskers are the 1.5X interquartile range, and individual points are outliers. **g** Row heatmaps showing differentially accessible peaks between Control and SOX9 MO treatments. Each line

in the heatmap is a peak. Peaks were initially ranked by log₂Fold Change (log₂FC) (Control vs. SOX9 MO) and vertical lines were drawn at a log₂FC threshold of 0.5 in both directions. Differential accessibility analysis was performed using the DBA_EDGER method within the R package DiffBind. **h** Plot showing the overlap of peaks from (**g**) with SOX9 and PanKla CUT&RUN peaks as well as ATAC-seq peaks at promoter regions. Vertical black lines indicate overlap. SOX9 and PanKla peaks are enriched among the significant (FDR < 0.05) peaks that are depleted in the SOX9 MO transfected NCCs whereas promoter peaks don't show a strong enrichment in the same direction. A chi-squared test was used to test for the association of lactylation or SOX9 peak status and whether that peak is enriched or depleted in the SOX9 MO condition (SOX9 $\chi^2$ = 2619.7, df = 1; PanKla $\chi^2$ = 1825.7, df = 1; Promoter $\chi^2$ = 57.8, df = 1). **i** Diagram depicting experimental strategy for gain-of-function experiment involving the injection of HH4 embryos with either an *pCI-H2B-RFP* construct or a mixture of *SOX9* and *TEA1-VPR* constructs. Embryos were collected at HH6, dissected to enrich for NPBCs and subjected to CUT&RUN for PanKla. **j** Profile plot showing the average cumulative TMM normalized HH6 PanKla CUT&RUN signal of RFP (magenta) and SOX9 + TEA1-VPR (purple) samples at HH9 PanKla CUT&RUN consensus peakset. **k** Genome browser tracks showing replicate average TMM normalized PanKla signal (HH6) for RFP and over-expression (OE) samples at the *SOX10* genomic locus. PanKla CUT&RUN tracks from HH9 samples is also shown. RefSeq gene annotation tracks are used to visualize genes in genome browser panels. Non-curated non-coding RNA annotations are not displayed.

Our results suggest that lactylated and acetylated (H3K27ac) genomic regions are subject to differential regulation during NCC development. These findings are consistent with the dynamics of these marks in different experimental systems. In their 2019 study, Zhang et al. show that the dynamics of lactylation and acetylation in macrophages differ[10]. The authors show that macrophage polarization

results in a significant increase in H3K18La levels and a concomitant decrease in H3K18ac levels. We also find that the deposition patterns of lactylation and acetylation differ in the genome of NCCs, and that lactylated *cis*-regulatory elements may or may not contain acetylation marks (such as H3K27ac). Since genomic loci enriched for lactylation marks are highly accessible and associated with active genes, it is

possible that they also act as enhancers alongside regions that contain both modifications.

The implication of SOX9 and YAP/TEAD in facilitating the cell-type-specific deposition of lactyl-CoA is also significant given what is known about the function of these factors in NCC development. Previous studies have shown that increased glycolytic flux in NCCs leads to the stabilization of YAP/TEAD, which go on to occupy NCC-specific enhancers[4,37]. Among the SOXE factors important for NCC development, SOX9 is specifically enriched in these cells with an expression pattern, in the stages spanning NCC specification and migration, that recapitulates that of lactylation levels[37]. Furthermore, SOX9 is known to interact with transcriptional co-activator and (recently described) lactyltransferase p300 to drive cell-type-specific gene expression[38,39]. In a 2018 study, Liu et al. used chondrogenesis as a model to show, among other findings, that SOX9 is involved in removing repressive marks and establishing active-promoter and active-enhancer marks at cartilage-specific loci[40]. Indeed, we found that SOX9 and YAP/TEAD were both significant predictors of PanKla deposition (i.e., lactylation) when the relationship between the three variables is modeled using multiple regression. Moreover, SOX9 loss-of-function lead to a significant loss in the accessibility of lactylated genomic regions while SOX9 and TEA1-VPR over-expression resulted in a global increase of PanKla signal when the embryos were assayed at HH6 (a stage when *SOX9* is not expressed). These results implicate SOX9 and YAP/TEAD in a mechanism underlying NCC-specific deposition of lactylation marks.

The identification of lactylation as part of a mechanism that couples the metabolic state of NCCs to GRN deployment and developmental gene expression has important clinical implications. Neurocristopathies, such as orofacial clefts or congenital heart defects, account for one third of all congenital malformations (CMs)[41,42]. Despite important advancements made in characterizing the genetic factors linked to neurocristopathies, we still lack a detailed understanding of the mechanisms through which non-genetic factors contribute to the multifactorial etiology of these conditions. This discrepancy is perhaps best brought to light when considering the results of recent large-scale genomic studies of CM probands. For example, in a 2018 study, Jin et al. show that a large fraction of congenital heart defect cases (including those caused by NCC dysfunction) could not be explained by genetic factors, thereby highlighting the importance of non-genetic contributions to the etiology and pathogenesis of these CMs[43]. Environmental glucose is a non-genetic factor that contributes to the etiology of CMs through its effect on the metabolic state of embryonic cell types. Indeed, maternal metabolic disorders that alter glucose levels in the developing embryo, such as pre-gestational Maternal Diabetes Mellitus (MDM), significantly increase the risk of neurocristopathies and other CMs[44–46]. Our findings suggest that metabolic perturbations during embryogenesis may, among other things, disrupt a basal-metabolism-derived epigenomic signature. This could ultimately contribute to aberrant GRN deployment and interfere with proper development of embryonic cell types. An enhanced understanding of the environmental aspects underlying the etiology of congenital malformations may serve as a platform to explore potential therapeutic interventions.

## Methods

### Chick embryo collection and electroporation
Fertilized White Leghorn eggs were purchased from the University of Connecticut (Department of Animal Sciences). The eggs were incubated at 37 °C until the embryos reached the desired developmental stage. Embryos were collected and cultured following the EC protocol[47]. Once collected, the Hamburger and Hamilton staging system[14] was used to stage and sort embryos. To transfect embryos, morpholinos or DNA expression vectors were injected in the space between the epiblast and the vitelline membrane of dissected embryos at HH4 and electroporated with platinum electrodes by passing five 50

msec pulses of 5.2 V[18]. Transfected embryos were cultured in 800 ul of thin egg albumin in 35 mm petri plates in a 37 °C humidified incubator. The embryos were incubated until they reached the desired stage and processed based on the specific experiment.

### NCC explant culture system
Prior to embryo dissection, tissue cultureware (Celltreat, 229168-chambered microscope slides were used for imaging) were coated with 25 ug/ml fibronectin (Sigma, F1141) and incubated for at least 45 min in a 37 °C humidified incubator maintained at 5% $CO_2$ atmosphere. After incubation, the fibronectin solution was removed from the cultureware and DMEM (Sigma, D6046) supplemented with 10% FBS (Thermo Fisher Scientific, 10-437-028) was added. NCC explants were obtained by thinly micro-dissecting dorsal neural folds from HH9 (6–7 somite) embryos in Ringer's solution as previously described[18]. Each dissected neural fold was then aspirated in a minimal amount of Ringer's solution (2.5 ul) and placed in a well of the fibronectin-coated tissue cultureware containing media. The explants were incubated for at least 12 h under normoxic conditions in a humidified 37 °C incubator maintained at 5% $CO_2$ atmosphere. The duration of incubation was determined by the appropriate conditions pertaining to specific experiments.

### IF staining of explant cultures
Upon conclusion of culture, explants were placed on wet ice, rinsed once with ice-cold 1xPBS, and fixed with a series of ice-cold formaldehyde (FA) solutions (Polysciences Inc., 18814-10) in 1X PBS. The fixation series was as follows: 0.5% FA for 5 min, 1% FA for 5 min, 1.5% FA for 10 min, finally 2% FA for 10 min. The explants were then washed once for 5 min with 1X PBS and permeabilized by incubating with a 0.1% NP-40 solution (Sigma, I8896) (in 1X PBS) at 37 °C for 30 min in a humidified incubator. The explants were washed twice for 5 min with 1X PBS at room temperature. Blocking was performed by incubating explants with 1% BSA (Roche, 03117332001) in 1X PBS for 30 min at 37 °C in a humidified incubator. After blocking, primary antibodies (diluted in blocking solution) were added, and the explants were incubated for 1 h at 37 °C in a humidified incubator. The explants were washed twice for 10 min in 1X PBS at room temperature with gentle rocking. After washing, secondary antibodies (diluted in blocking solution) were added to the explants and the samples were incubated for 45 min at 37 °C in a humidified incubator. The explants were washed three times for 10 min in 1X PBS with gentle rocking while covered. Finally, the explants were incubated with DAPI (300 nM in 1X PBS, Thermo Fisher Scientific, D1306) and Alexa647-Phalloidin (1:250 in 1X PBS, Thermo Fisher Scientific, A22287) for 10 min at room temperature. The explants were washed two times for 5 min in 1X PBS, mounted in Fluoromount (Electron Microscopy Sciences, 17984-25), and imaged on a Zeiss Imager.Z2 fluorescent microscope. The antibodies used for immunohistochemistry experiments are as follows: rabbit polyclonal anti-L-Lactyl-lysine (PTM Biolabs, PTM-1401 polyclonal) (1:200), Donkey anti-Rabbit IgG (H + L) Secondary Antibody, Alexa Fluor 488 (Thermo Fisher Scientific, A21206) (1:1000).

### IF staining of whole embryos
Embryos were collected in Ringer's solution on filter paper as previously described[47] and fixed in 4% PFA (in 1X PBS) for 20 min at room temperature with gentle rocking. After fixation, the embryos were transferred to a 1X TBST (1X TBS with 0.1% TritonX-100 (Sigma, X-100)) and dissected from the filter paper. The embryos were then washed 3 times for 10 min each in 1X TBTD (1X TBST with 1% DMSO). The wash solution was then replaced with blocking solution (1X TBTD with 10% Donkey Serum (Equitech-Bio, sd300500)) and the embryos were incubated for 1 h at room temperature while rocking. After blocking, primary antibodies (diluted in blocking solution) were added, and

embryos were incubated for at 4 °C overnight. The next day, the embryos were washed four times for 15 min each in TBTD while rocking. After washing, embryos were blocked in blocking solution for 30 min and fluorophore-conjugated secondary antibodies (in blocking solution) were added to the samples. Embryos were incubated in with secondary antibodies for 2 h at room temperature and then washed four time for 15 min each at room temperature while rocking and covered. Finally, DAPI (Thermo Fisher Scientific, D1306) was added to the embryos (diluted to 300 nM in 1X TBTD) and the embryos were incubated for 10 min at room temperature. The embryos were washed three times for 5 min in 1X TBST and imaged on a Zeiss Imager.Z2 fluorescent microscope. The antibodies used for immunohistochemistry experiments are as follows: rabbit polyclonal anti-L-Lactyl-lysine (PTM Biolabs, PTM-1401 polyclonal) (1:200), mouse monoclonal anti-AP2 beta (Santa Cruz Biotechnology, sc-390119) (1:200), rabbit monoclonal anti-Sox9 (Millipore, AB5535) (1:200), rabbit monoclonal anti-Caspase3 (R&D Systems, AF835) (1:250), rabbit monoclonal anti-pH3 (Cell Signaling, 9701S) (1:250), Donkey anti-Rabbit IgG (H + L) Secondary Antibody, Alexa Fluor 488 (Thermo Fisher Scientific, A21206) (1:1000), Goat anti-Mouse IgG1 (H + L) Secondary Antibody, Alexa Fluor 633 (Thermo Fisher Scientific, A21126) (1:1000), and Alexa Fluor 647-conjugated AffiniPure Fab Fragment Donkey Anti-Rabbit IgG (H + L) (Jackson ImmunoResearch, 711-607-003) (1:1000).

### Cryosectioning of embryonic tissue and IF staining on slides

Embryos were collected on filter paper in Ringer's solution and fixed in 4% PFA for 30 min at room temperature. The embryos were dissected from the filter paper in 1X TBS-Tween (1X TBS with 0.1% Tween-20 (Sigma, P1379)) and washed in 1X TBS-Tween three times for 10 min. After washing the embryos were transferred to 5% sucrose (Sigma, S7903) in 1X PBS and incubated at room temperature for 2 h while rocking. Next, the embryos were transferred to 15% sucrose (in 1X PBS) and incubated overnight at 4 °C. The next day, the 15% sucrose solution was aspirated carefully, and a 7.5% gelatin (Sigma, G1890) solution (at 37 °C) prepared in 15% sucrose was added to the embryos. The embryos were incubated at 37 °C in gelatin for 2 h and embedded in silicone molds which were flash frozen in liquid nitrogen once the gelatin had solidified. The tissue in the frozen gelatin block was sectioned using a CryoStar cryotome and the sections were transferred on charged glass slides (VWR, 48311-703). The slides were left to dry overnight at room temperature. The next day, the slides were incubated in 1X TBS-Tween heated to 42 °C for 10 min and washed in 1X TBS-Tween three times for 10 min. A hydrophobic pen was used to draw a border around the sectioned tissue on the slides and immunohistochemistry was performed suing the same steps as previously stated with embryonic tissue with the only difference being the use of 1X TBS-Tween in place of 1X TBST and 1X TBTD solutions. After incubating the slides with DAPI, they were mounted with Fluoromount (Electron Microscopy Sciences, 17984-25) using a glass coverslip and imaged on a Zeiss Imager.Z2 fluorescent microscope.

### Cell suspension staining of embryonic tissue for flow cytometry

Embryonic tissue was micro-dissected from wild-type embryos in Ringer's solution. HH6, HH9, and HH12–13 embryos were dissected as follows: five whole HH6 embryos were dissected including tissue surrounding the neural plate border, 13 HH9 embryonic heads were dissected until the first somite, and 11 HH12–13 embryonic heads were dissected until the first somite. Two biological replicates containing the same number of WEs (for HH6) or heads (for HH9 and HH12–13) were collected. Once dissected the embryonic tissue was rinsed once in Accumax (Innovative Cell Technologies, AM105) solution and allowed to settle to the bottom of the tubes. The supernatant was removed and fresh Accumax solution was added to the samples. The embryonic tissue was allowed to dissociate at room temperature for 30 min (with gentle pipetting every 10 min) to achieve a single cell

suspension. The dissociated cells were spun at 300 rcf for 5 min at room temperature (these centrifugation conditions were kept consistent for the remainder of the protocol). The supernatant was removed, and the cells were rinsed with 1X DPBS (without calcium and magnesium, Gibco, 14190-144). The samples were spun down and 1X DPBS was added to resuspend the cells. An equal volume of 4% PFA (in 1X PBS) was added to the cells to achieve a 2% PFA solution. The samples were mixed by pipetting gently up and down and fixed at room temperature for 15 min. After fixation, the cells were rinsed in 1X DPBS and spun down. The supernatant was discarded, and the cells were permeabilized by adding 0.7% Tween-20 (in 1X PBS) and incubating for 15 min at room temperature. The cells were spun down and then rinsed once in flow buffer (1% BSA in 1X DPBS). After spinning down and removing the supernatant, the cells were washed once for 30 min in blocking buffer (1%BSA, 10% donkey serum (Equitech-Bio, sd300500), and 0.5% Tween-20 (in 1X PBS)) at room temperature. The samples were spun down and primary antibodies were added to the cells at a 1:100 dilution in blocking buffer and samples were incubated at room temperature for 1 h (in the dark). After incubation with primary antibodies, the cells were rinsed twice with flow buffer, spinning down after every time and removing as much of the supernatant as possible without disturbing the cell pellet. Secondary antibodies were added to the cells at a 1:1000 dilution in blocking buffer and samples were incubated at room temperature for 30 min. The samples were rinsed three times with flow buffer spinning down after every time. Secondary antibody control samples were incubated only with secondary antibodies. The antibodies used for flow cytometry experiments are as follows: rabbit polyclonal anti-L-Lactyl-lysine (PTM Biolabs, PTM-1401 polyclonal) (1:100), mouse monoclonal anti-AP2 beta (Santa Cruz Biotechnology, sc-390119) (1:100), mouse monoclonal anti-PAX7 (Developmental Studies Hybridoma Bank, AB_528428) (1:100), mouse monoclonal alpha-Tubulin (Cell Signaling Technology, #3873), Goat anti-Mouse IgG1 Cross Adsorbed Secondary Antibody (Alexa Fluor 488, Thermo Fisher Scientific, A21121) (1:1000), and Alexa Fluor 647-AffiniPure Fab Fragment Donkey Anti-Rabbit IgG (H + L) (Jackson ImmunoResearch, 711-607-003) (1:1000). The samples were analyzed on a BD FACSCelesta flow cytometer, measuring Alexa647 and Alexa488 fluorescence intensity of each sample including single color controls (samples positive for either Alexa488 or Alexa568), background signal controls (samples stained with a combination of Alexa568/Alexa488 secondary antibodies but no primary antibodies), and unstained controls.

### Enhancer reporter cloning

Putative enhancer elements were amplified from genomic DNA extracted from HH9 chicken embryos with sequence-specific primers containing overhangs compatible with the pTK-EGFP using the Q5® High-Fidelity 2X Master Mix (NEB, M0492S). Amplified fragments were gel extracted and purified using the Wizard® SV Gel and PCR Clean-up System (Promega, A9282) and cloned in SmaI-digested pTK-EGFP using the NEBuilder® HiFi DNA Assembly Master Mix (NEB, E2621). The galGal6 genomic positions for SNAI2 E1, SEMA3D E2, and SEMA3D E1 are chr2:108412500–108413620, chr1:8854387–8855152, and chr1:8852988–8853832 respectively. The following primers were used for amplification:

SNAI2_E1_FW 5′AGCTCTTACGCGTGCTAGCCCTAGTTCTCAGTTACACTGGTGTG3′,
SNAI2_E1_REV 5′TAGATCGCAGATCTCGAGCCCAGCAAATGCTCTGCTCCTG3′,
SEMA3D_E1_FW 5′AGCTCTTACGCGTGCTAGCCCGTGACACCTAAAAATCCAACCTGG3′,
SEMA3D_E1_REV 5′TAGATCGCAGATCTCGAGCCCAAGGCATTTTTCCCCTTAAGATCAGG3′,
SEMA3D_E2_FW 5′AGCTCTTACGCGTGCTAGCCCATGCTGGTCTCTGAGGAAAGC3′,

SEMA3D_E2_REV  5′TAGATCGCAGATCTCGAGCCCTCACATCCTG
ATCTGAGCCAG3′.

## SOX9 over-expression vector cloning
The chicken *SOX9* coding sequence (CDS) was amplified from cDNA using CDS-specific primers designed to contain overhangs that overlap with the 5′ and 3′ of a PmeI-digested pCI-H2B-RFP vector (FW 5′GCGCCTTAATTAACGTTTGCCACCATGGACTACAAGGACGACGACG ACAAGCCGCTTTCTCGCATGAATCT3′; REV 5′CCATTTGCATGCATGT TTGCTTTAAGGCCGGGTGAG3′) as well as a FLAG tag and Kozak sequences on the 5′ end. The pCI-H2B-RFP vector was digested with PmeI, separated on an agarose gel, and purified using the Wizard(R) SV Gel and PCR Clean-Up System (Promega, A9282). The linearized pCI-H2B-RFP plasmid and *SOX9* coding sequence were subjected to a two fragment Gibson assembly with the NEBuilder(R) HiFi DNA Assembly Master Mix (NEB, E2621).

## LDHA/B loss-of-function effects on NCC migration
LDHA and LDHB knockdown was achieved using two FITC-labeled translation blocking MOs (GeneTools) targeted to the mRNA of each gene:

LDHA MO−5′ATGATCCTTGAGAGACATGGTGTAC3′,

LDHB MO−5′CTCCTTCAGGGTCGCCATAACGTCC3′. To assess the effects of LDHA/B knock-down on NCC migration, a mixture of 0.75 mM each of LDHA and LDHB MOs was injected on the right side of the embryo, whereas 1 mM of a GTBlue-Control MO (GeneTools) was injected on the left side. In each case pTK-mCherry-Tfap2a-E1[18] was co-injected as carrier DNA to a final concentration of 1 ug/ul. Morpholino solutions were prepared in 10 mM Tris-HCl pH 8.0. Embryos were developed until HH9+/10 and screened for the correct stage of NCC migration using the *TFAP2A* enhancer. Once the correct stage was reached, the embryos were collected and fixed in 4% PFA (Sigma, P6148) and stained as described above for TFAP2B to mark migrating cranial NCCs.

## LDHA/B MO validation
LDHA and LDHB sequences containing the exon(s) targeted by the translation-blocking MO (LDHA−5′ CAGTACACCATGTCTCTCAAGG ATCATCTCATCCACAATGTCCACAAAGAGGAGCACGCTCATGCCCAC AACAAGATCAGCGTGGTTGGTGTGGGTGCAGTTGGAATGGCTTGTG CCATCAGCATCCTGATGAAG3′ and LDHB−5′ AATGCGGATTGAACGC GGAGGCCGCCGCCTCCCGTCGACGCTGTGCCCGCCCCGAGGGCGC GGTGAGGGGCTCCTCCACGCATCCCAGCCCGCCCCTTCCGCTGCGGA GCGCAGATTTCCCGAGCCCCGCCACGGTCACGGTACTGCTCCCGGT TCTCCTTTCACCGCACCGATCCGGACGTTATGGCGACCCTGAAGGAG AAGCTGATCACCCCCGTGGCCGCGGGCAGCACGGTTCCCAGCAACAA GATCACCGTGGTGGGGGTCGGGCAGGTGGGGATGGCGTGTGCCATC AGCATCCTCGGCAAG3′) were designed in silico to include a 5′ over-hang overlapping with a PmeI-digested pCI-H2B-RFP vector and a 3′ overhang overlapping with the mRFP1 sequence. The sequences were then ordered as double-stranded DNA from IDT (gBlocks® Gene Fragments). The mRFP1 sequence was amplified from the pCI-H2B-RFP vector with a forward primer designed against the 5′ end of the sequence and a reverse primer designed against the 3′ end of the *RFP* sequence but also containing an overhang that overlaps with a NotI-digested pCI-H2B-RFP vector. The pCI-H2B-RFP vector was then double-digested by PmeI and NotI removing the *RFP* sequence entirely. Finally, the *LDHA* or *LDHB* sequence with the 5′ PmeI overhang and 3′ RFP overhang, *RFP* sequence with the 3′ NotI overhang, and PmeI/NotI double-digested pCI-H2B-RFP were subjected to a three fragment Gibson assembly with the NEBuilder® HiFi DNA Assembly Master Mix (NEB, E2621) resulting in the formation of morpholino test constructs that contain the LDHA or LDHB exon sequence fused, in frame, to mRFP1. Each test construct (1 ug/ul) was co-injected with either GTBlue-Control or LDHA/B MOs and RFP expression was used as a measure of MO knockdown efficiency.

## LDHA/B MO effects on lactylation levels
Individual embryos were injected at HH4 with LDHA/B MO mix (0.75 mM each LDHA and LDHB) on one half and FITC-Control MO (0.75 mM) on the other half as previously stated. A non-fluorescent construct (pTK-Luciferase) was used as carrier DNA. The embryos were allowed to develop until HH9 at which point they were collected in Ringer's solution and half of the head was dissected and assigned to control or treated samples The samples were subjected to the cell suspension staining protocol described above while being protected from light (*n* = 12 half embryonic heads for both FITC-control and LDHA/B MO samples). The antibodies used for this flow cytometry experiment are as follows: rabbit polyclonal anti-L-Lactyl-lysine (PTM Biolabs, PTM-1401 polyclonal) (1:100) and Alexa Fluor 647-conjugated AffiniPure Fab Fragment Donkey Anti-Rabbit IgG (H + L) (Jackson ImmunoResearch, 711-607-003) (1:1000). The samples were analyzed on a BD FACSCelesta flow cytometer, measuring Alexa647 and FITC fluorescence intensity of each sample.

## LDHA/B MO and Control MO Nanostring experiment
Embryos were injected with LDHA/B and GTBlue-Control MOs in a paired fashion as describe above. At HH9, the embryos were collected in Ringer's solution and neural folds from control and experimental sides of three embryos were micro-dissected and placed in 10 ul of Cells-to-CT Lysis Buffer (Thermo Fisher Scientific, 4383573). The samples were incubated for 5 min on ice and flash frozen in liquid nitrogen for storage. The samples were ultimately subjected to the protocol for analysis on the Nanostring MAX/FLEX system with a probe set containing 200 probes following the manufacturer's protocol.

## Lactate effects on NCC migration
Neural folds were dissected from HH9 embryos and cultured as described previously in paired fashion (*n* = 14 biological replicates) in media containing 1X PBS or 10 mM sodium lactate (Sigma, 71718). Explants were cultured for 48 h using previously stated settings. Upon conclusion of culture, they were placed on wet ice, rinsed once with ice-cold 1xPBS, and fixed with a series of ice-cold solutions containing both glutaraldehyde (GA) (Sigma, G5882) and FA in 1X PBS. The fixation series was as follows: 0.25% FA and 0.25% GA for 5 min, 0.5% FA and 0.25% GA for 5 min, finally 1% FA and 0.25% GA for 10 min. The explants were then washed once for 5 min with 1X PBS and incubated with DAPI (1:1000 in 1X PBS) and Alexa647-Phalloidin (1:250 in 1X PBS) for 10 min at room temperature. The explants were washed once with 1X PBS for 5 min and mounted using Fluoromount.

## CUT&RUN
NCCs and NBPCs were obtained by micro-dissecting dorsal neural folds from HH9 embryos (*n* = 5 embryos, *n* = 10 neural folds) and NPB tissue from HH6 embryos (*n* = 2–3 embryos), respectively. PSM tissue was obtained by dissecting the posterior region of HH9 embryos (*n* = 5 embryos, *n* = 10 PSM strips) as depicted in Fig. 4a. A cell suspension was achieved by dissociating embryonic tissue with Accumax (Innovative Cell Technologies, AM105) for 30 min at room temperature under mild agitation and gentle pipetting halfway through the incubation time. The cells were then subjected to the low-salt CUT&RUN protocol[28]. Briefly, cells were immobilized on BioMag Plus Concavilin A magnetic beads (Bangs Laboratories, BP531) and incubated with rabbit polyclonal anti-L-Lactyl-lysine (PTM Biolabs, PTM-1401 polyclonal) (1:50), rabbit polyclonal anti-H3K18La (PTM Biolabs, PTM-1406 polyclonal) (1:50), or rabbit monoclonal anti-Sox9 (Millipore Sigma, AB5535) (1:50) antibodies overnight at 4 °C. After washing unbound antibody, protein AG-MNase was added to the cells to a final concentration of 700 ng/ml and the samples were incubated for 1 h at 4 °C. After rinsing away unbound protein AG-MNase, the cells were incubated in the low-salt incubation buffer (containing 10 mM CaCl$_2$ to activate the MNase enzyme) for 30 min at 0 °C (on wet ice). MNase

digestion was terminated by removing the incubation buffer and adding the STOP buffer containing 20 mM EGTA. CUT&RUN fragments were released from the insoluble nuclear chromatin by incubating samples at 37 °C for 30 min. DNA was isolated via phenol-chloroform extraction and ethanol precipitation. Critical to the success of this protocol was performing all initial cell centrifugation steps (prior to immobilization of Concanavalin A beads) in a swinging bucket centrifuge equipped with adapters for 1.5 ml microfuge tubes to minimize cell loss. The final Digitonin (Millipore, 300410) concentration used for all CUT&RUN experiments in this study was 0.05%. Carryover E. coli genomic DNA in the protein AG-MNase sample was used to perform spike-in normalization wherever applicable[28]. Protein AG-MNase was kindly provided by Dr. Steven Henikoff.

### CUT&RUN library preparation

DNA libraries for CUT&RUN experiments were performed using the NEBNext Ultra II DNA Library Prep Kit (NEB, E7645) following the manufacturer's protocol. Quality control of prepared libraries was performed using an Agilent 4200 Tapestation with D5000 Reagents (Agilent, Part# 5067-5589) and D500 ScreenTape (Agilent, Part# 5067-5588). Libraires were pooled to equimolar concentrations using the NEB Library Quantification Kit (NEB E730S) and sequenced with paired-end 37-bp reads on an Illumina NextSeq 500 instrument.

### ATAC-seq

Neural folds were dissected from HH9 embryos as described in the CUT&RUN protocol ($n = 2$–$4$ embryos, $n = 4$–$8$ neural folds) and dissociated in Accumax (Innovative Cell Technologies, AM105) for 30 min at room temperature to achieve a single cell suspension. ATAC-seq was performed as previously described[48]. Briefly, dissociated cells were collected by centrifugation at 500 rcf for 5 min in a swinging bucket centrifuge and washed in resuspension buffer containing 10 mM Tris-HCl pH 7.4, 10 mM NaCl, and 3 mM $MgCl_2$. After washing, the cells were resuspended in lysis buffer composed of the resuspension buffer with 0.1% NP-40 (Sigma, I8896), 0.1% Tween-20 (Sigma, P1379), and 0.01% Digitonin (Millipore, 300410). The samples were incubated on ice for 3 min, at which point the lysis buffer was washed out with resuspension buffer containing only 0.1% Tween-20. Upon centrifugation at 500 rcf for 5 min, the supernatant was aspirated, and nuclei were resuspended in 50 ul of transposition mixture containing 10 mM Tris-HCl pH 7.4, 5 mM $MgCl_2$, 10% Dimethyl Formamide (Sigma, D4551), 1X PBS, 0.1% Tween-20, 0.01% Digitonin, and 5 ul Tn5 transposase (Illumina, 20034210). Transposition was performed for 1 h at 37 °C in a thermal mixer. The pre-amplification DNA was purified using the Qiagen Minelute Reaction Cleanup Kit (Qiagen, 28204). Library amplification PCR was performed using the Q5® High-Fidelity 2X Master Mix (NEB, M0492S). Library size selection and cleanup was carried out using Ampure XP beads (Beckman Coulter, A63881). Prepared libraries were run on an Agilent 4200 Tapestation with D5000 Reagents and D500 ScreenTape for quality control and final libraires were pooled to equimolar concentrations using the NEB Library Quantification Kit. Sequencing was performed on an Illumina NextSeq 500 instrument with paired-end 37-bp reads.

### scATAC-seq—tissue isolation and library preparation

Dorsal neural folds were manually dissected from HH9 embryos in two biological replicates ($n = 10$–$20$ NFs/replicate). After tissue collection, cells were dissociated in Accumax (Innovative Cell Technologies, AM105) for 20 min at RT under mild agitation and nuclei preparation was performed according to the 10X Genomics protocol. Briefly, cells were lysed in cold lysis buffer containing 10 mM Tis-HCl pH7.4, 10 mM NaCl, 3 mM $MgCl_2$, 1% BSA, 0.01% Tween 20, 0.01% Nonidet P40 Substitute, and 0.001% Digitonin for 5 min on ice. Following cell lysis, cells were washed in 10 mM Tis-HCl pH7.4, 10 mM NaCl, 3 mM MgCl2, 1% BSA, and 0.1% Tween 20 and resuspended in 1x Nuclei Buffer (10x

Genomics, 2000153/2000207) at a concentration of 3500 nuclei per ul. Nuclei were then immediately used to generate scATAC-seq libraries via the Chromium Single Cell ATAC pipeline (10x Genomics, PN-1000175).

### (R)-GNE-140/DMSO ATAC-seq experiment

Neural folds were dissected from six HH9 embryos and cultured as described previously in paired fashion in media containing DMSO or 40 μM (R)-GNE-140 (Selleck Chemicals, S6675) ($n = 6$ neural folds per control/treatment). This experiment was performed with two biological replicates. Explants were cultured for 12 h using previously stated settings, at which point they were collected from the plate by incubation with Accumax (Innovative Cell Technologies, AM105) for 10 min at 37 °C. The single cells from each condition/sample were subjected to the ATAC-seq protocol (described above).

### SOX9/Control MO ATAC-seq experiment

*SOX9* knock-down was achieved using a translation-blocking MO targeted to the mRNA of the gene[25]. Whole embryos were injected with either the SOX9 MO or the GTBlue-Control MO at a final concentration of 1 mM in Tris-HCl pH 8.0 supplemented with 1 ug/ul carrier DNA. Embryos were cultured until HH9 (6–7 somites) as described previously and transferred to Ringer's where the dorsal neural folds of SOX9 MO or GTBlue-Control MO transfected embryos were microdissected ($n = 2$ embryos, $n = 4$ neural folds per biological replicate) and subjected to the ATAC-seq protocol (described above).

### SOX9/TEA1-VPR GOF CUT&RUN experiment

Whole embryos were electroporated at HH4 with either 2.5 μg/μl of pCI-H2B-RFP (control condition)[4] or a mixture containing 1.25 μg/μl pCI-SOX9-H2B-RFP (this paper) and 1.25 μg/μl TEA1-VPR[4]. Both control and experimental (SOX9/TEA1-VPR) embryos were cultured in albumen. Embryos were collected in Ringer's solution when they reached HH6 and were dissected to enrich for neural plate border cells as previously described[18]. CUT&RUN for PanKla ($n = 2$ embryos/sample) was performed on control and experimental samples as described above.

## Quantification and statistical analysis
### Flow cytometry data analysis

Initial flow cytometry data analysis was performed using the FCS Express 7 software. Events were first gated by SSC-A vs. FSC-A to remove cellular debris and select the population of embryonic head cells. Next, the events were sub-gated by FSC-H vs. FSC-A to select single cells. Finally, the single cells for the control samples (background and unstained controls) were plotted based on the analyzed fluorescent channels to set the gates for the signal in each channel. Experimental samples were analyzed to identify populations of interest (PAX7-positive and TFAP2B-positive cells that were also positive for PanKla). The florescence intensity (area) data for each analyzed fluorophore in the experimental samples pertaining to the analysis of lactylation levels during NCC development were imported into R and visualized. Statistical analysis of lactylation levels in HH6, HH9, and HH12–13 embryos included performing a Kruskal–Wallis test, followed by ad hoc Wilcoxon rank sum test with FDR correction used for *p* values. For the experiment involving FITC-control and LDHA/B MO transfected embryonic heads, the analysis of lactylation levels in FITC-positive cells was primarily done in FCS Express by plotting lactylation levels of cells as stacked histograms.

### Quantification of NCC migration in MO-transfected embryos

Embryos were imaged on a Zeiss Imager.Z2 fluorescent microscope focusing on the plane of migrating (TFAP2B+) NCCs on top of the embryo. The CZI image files were then opened in FIJI[49] and the tracking tool was used to track the NCC nuclei on each side of the embryo and quantify the area of migration.

### Caspase3 and pH3 quantifications in MO-transfected embryos

LDHA/B MO and Control MO transfected embryos were stained for Caspase3/TFAP2B or pH3/TFAP2B as described above. The embryos were then imaged on a Zeiss Imager.Z2 fluorescent microscope focusing on the plane of migrating (TFAP2B+) NCCs on top of the embryo. The CZI image files were opened in FIJI and Casp3 ($n = 8$ embryos) or pH3 ($n = 6$ embryos) fluorescence intensity was quantified in the domains of migrating NCC in the embryonic head (midbrain region) while also including the surrounding lateral ectoderm tissue. The fluorescent signal was normalized to the area of quantification (by dividing signal by area) and the values were reported.

### Nanostring data analysis

Nanostring data was analyzed using the nSolver software package (Nanostring Technologies) using the geometric mean of the counts for a set of positive/negative control probes for background subtraction and normalization. After normalization, the gene counts for Control and LDHA/B MO samples were used to calculate the fold change for each of the three biological replicates ((experimental counts − control counts)/control counts). The fold changes were averaged and plotted with standard deviation used as a measure of dispersion.

### Quantification of NCC migration in explant cultures

Explants were imaged on a Zeiss Zoom.V16 fluorescent stereoscope focusing on the plane of migrating NCCs at the top of the embryo The CZI image files were then opened in FIJI and the tracking tool was used to track the NCC nuclei on each side of the embryo and quantify the area of migration. Higher magnification images of explant cells were captured using a Zeiss Imager.Z2 fluorescent microscope.

### CUT&RUN data analysis

Processing the CUT&RUN data involved trimming Illumina adapter sequences from the paired-end reads using Cutadapt (v2.10)[50] and selecting reads that were at least 25-bp long using the following arguments `-a AGATCGGAAGAGCACACGTCTGAACTCCAGTCA -A AGATCGGAAGAGCGTCGTGTAGGGAAAGAGTGT --minimum-length=25 -j 0`. The reads were then aligned to the reference chicken galGal6 assembly using Bowtie2 (v2.4.2)[51] with the following arguments "`--local --very-sensitive-local --no-unal --no-mixed --no-discordant -I 10 -X 1000`". Duplicate reads were then marked by the Picard MarkDuplicates tool (https://github.com/broadinstitute/picard) and BAM files were filtered with SAMtools to discard unmapped reads, reads which were not the primary alignment, reads failing platform/vendor quality checks, and PCR/optical duplicates (`-F 1804 -f 2`). MACS2[52] was used to call peaks with a q value cutoff equal to 0.05 with the following arguments "`-f BAMPE -g 1218492533 -q 0.05 call-summits`". Consensus peaksets between biological replicates for each assayed histone mark or factor were determined by intersecting the peaksets of each replicate by using the BEDTools[53] intersect function and only keeping peaks that were called in both replicates. Peaks corresponding to the IgG control CUT&RUN[18] were subtracted from each consensus peakset using the BEDTools subtract function. The subread package function featureCounts[54] was used to generate peak count matrices for specific BAM files using the following arguments "`--fracOverlap 0.5 --minOverlap 5 -p`". Before analyzing peak matrices, the R package edgeR[55] was employed to first obtain TMM-normalized effective library sizes that were then used to normalize raw peak counts with the `cpm()` function. Peaks were annotated based on their closest gene using the R package ChIPSeeker[56] using Ensembl 99 for the galGal6 genome. Tornado and profile plots were generated using the python package DeepTools[57]. Motif enrichment was performed using HOMER by providing the consensus peakset, chicken genome fasta file, and specifying he argument `-size given`.

Reads per genomic content (RPGC) normalized BigWig files, for visualization of normalized read pile-up in a genome browser (IGV v2.13.0)[58], were generated with the bamCoverage function of DeeTools using the following arguments "`--outFileFormat bigwig --bin-Size 5 --numberOfProcessors 16 --normalizeUsing RPGC --effectiveGenomeSize 1218492533 --extendReads`".

### PanKla CUT&RUN integration with WE vs. NCC LRT results

The time-course RNA-seq data of WE and NC cells, used in the present study, were generated in a study by Hovland et al.[13] and are available on GEO (GSE163961). The LRT analysis of the data was replicated using the source code from the repository associated with the study (https://doi.org/10.5281/zenodo.7044924). Genes associated with PanKla peaks at HH9 were projected onto the volcano plot comparing WE and NCCs. To compare the lactylation levels between NCC-enriched genes and WE-enriched genes, the signal from all lactylation peaks associated with a given gene (determine though peak-gene assignment as described above) was added and analyzed.

### PanKla CUT&RUN integration with time-course ATAC-seq datasets

The bulk time-course ATAC-seq data of NC cells at HH6, HH8, and HH10 stages of development were generated in a study by Hovland et al. and are available on GEO (GSE163961). These datasets were used to determine the cumulative accessibility of PanKla peaks from HH9. Data were visualized using profile plots from DeepTools (as described above).

### NCC vs. PSM PanKla CUT&RUN differential analysis

The R package DiffBind (version 2.14.0) was used to analyze the differential deposition of lactylation in NC vs. PSM cells. To begin, prominent IgG-positive regions, determined from analyzing the IgG CUT&RUN in NCCs, were removed from the PanKla peaksets of both replicates for each NCC and PSM CUT&RUN. Peak files corresponding to each sample/replicate were then filtered to include only peaks that map to chromosomes (not contigs). Reads from NCC and PSM PanKla CUT&RUN BAM files were counted at a consensus peakset that contained peaks present in all samples/replicates. This was done by specifying `minOverlap=1` in the `dba.count()` function. Binding affinity matrix scores were reported after performing a TMM normalization using full library size. The comparison between samples was set up with the `dba.contrast()` function accounting for the fact that the samples were unpaired. Differential accessibility statistical analysis was performed using the `DBA_EDGER` method. Assignment of differential peaks to their closest gene was done using the R package ChIPseeker. The R package chromVAR was used to determine the variability in lactylation (between NCC and PSM samples) of sequences that contain known transcription factor motifs, in an effort to identify the motifs that are present among NCC- and PSM-enriched lactylation peaks. Vertebrate transcription factor binding profiles were obtained from JASPAR's CORE collection. A new DBA object was constructed as a summarized experiment from the existing NF vs. PSM differential analysis object and used as an input for the `computeDeviations()` function of chromVAR along with the JASPAR motifs matched to the BSgenome for galGal6. Motif deviation $z$-scores for each sample were obtained and the deviation $z$-scores of NCC or PSM replicates were averaged. Variability score of variable motifs was computed using the `computeVariability()` function of chromVAR.

### Multiple regression analysis of SOX9 and YAP1 occupancy as predictors of PanKla deposition to model the relationship of SOX9, YAP1, and PanKla

`featureCounts` was used to generate a peak count matrix (using the PanKla consensus peakset as a reference) for all BAM files (without duplicate reads) with previously stated specifications. The raw counts

of the peak count matrix were then normalized using edgeR with the TMM approach and reported as CPM values. Only peaks with CPM > 0 in all six samples (2 replicates each for SOX9, YAP1, and PanKla) were used for further analysis. This filtering resulted in 10,448 peaks out of the original total of 10,912 peaks and was performed to remove PanKla peaks that had inconsistent numbers of normalized reads between replicates of the samples being analyzed. The relationship between the variables was modeled using a multiple linear regression where SOX9 and YAP1 occupancy were set as explanatory variables and PanKla signal was set as a response variable. The `lm()` function in R was used to perform this analysis.

### Spike-in normalization and analysis of SOX9/TEA1-VPR GOF samples

PanKla CUT&RUN data from GOF experiment embryos was processed and analyzed initially as described above. To begin the spike-in normalization process, adapter trimmed PanKla CUT&RUN fastq files, corresponding to each sample/replicate, were aligned to the *E. coli* genome (GCA_001606525, assembly ASM160652v1) using Bowtie2 (v2.4.2) with the following arguments "`--local --very-sensitive-local --no-unal --no-mixed --no-discordant -I 10 -X 1000`". Duplicate reads were then marked by the Picard MarkDuplicates tool and BAM files were filtered with SAMtools to discard unmapped reads, reads which were not the primary alignment, reads failing platform/vendor quality checks, and PCR/optical duplicates (`-F 1804 -f 2`). The BAM files for both galGal6 and *E. coli* genome alignments were then converted to BED files with the `bedtools bamtobed` function. A new column containing the fragment length (of each position) was added to each BED file and the columns containing chromosome name (1st), start position (2nd column), end position (3rd column), and fragment length (last column) were extracted to generate new BED files (for both galGal6 and *E. coli* genome alignments corresponding to each sample/replicate). The C-shell script for spike-in calibration of CUT&RUN data, previously published by Skene et al.[28] was then used to obtain normalized bedgraph files for each sample/replicate with the scale argument, min_len argument, and max_len argument set to 10,000, 1, and 1000 respectively. The spike-in normalized bedgraph files were converted to BigWig format using the `bedGraphToBigWig` function and the average signal between replicates for each sample was calculated using the `wiggletools mean` function. Profile plots were generated using the python package DeepTools. The `computeMatrix` function was used in reference-point mode with the following arguments "`-bs 50, --skipZeros, --missingDataAsZero, --referencePoint center`". To generate the RFP aggregate peakset used as a control, first peaks were called using MACS2 with a q value cutoff equal to 0.05 with the following arguments "`-f BAMPE -g 1218492533 –nomodel -q 0.05 call-summits`". The peaks from the RFP replicate samples were then concatenated and merged to generate the final representative aggregate peakset which was used as control in the generation of profile plots as described above.

### TMM normalization and analysis of SOX9/TEA1-VPR GOF samples

To begin TMM normalization of the PanKla CUT&RUN data from the GOF experiment, a representative aggregate peakset was generated by first concatenating the called peaks (bed files) from all samples/replicates and then merging the peaks using the `bedtools merge` function. The `featureCounts` function was then used to generate a peak count matrix (using the aggregate peakset as a reference) for all BAM files (without duplicate reads). The peak count matrix was then imported into R and TMM normalization factors were calculated for each sample using the `calcNormFactors()` function of the edgeR package with the `method = "TMM"` argument specified. Size factors were calculated using the formula

Size Factor = TMM Factor*Library Size/1,000,000. The Library Size variable is defined as the total number of reads of a sample in the peak count matrix and differs between samples. The reciprocal of the size factors (1/Size Factor) was then obtained. The bamCoverage function was then used to generate TMM and library size normalized bigwig files with the following arguments specified `-bs 50 --scaleFactor {1/Size Factor}`. The TMM normalized bigwig files for each sample were then averaged using the wiggle tools mean function and profile plots were generated using the python package DeepTools as previously stated.

### ATAC-seq data analysis

For ATAC-seq data analysis, paired-end sequencing reads were trimmed using Cutadapt (v2.10) using the following arguments "`cutadapt -a CTGTCTCTTATACACATCT -A AGATGTGTATAAGAGACAG --minimum-length=25`". Bowtie2 (v2.4.2) was used to align trimmed reads to the reference chicken galGal6 assembly using the following arguments "`--no-unal --no-mixed --no-discordant -X 2000`". Duplicate reads were then marked by the Picard MarkDuplicates tool and SAMtools was used to filter BAM files to discard unmapped reads, reads which were not the primary alignment, reads failing platform/vendor quality checks, and PCR/optical duplicates (`-F 1804 -f 2`). MACS2 was used to call peaks genome-wide with a q value cutoff equal to 0.05 with the following arguments "`-f BAMPE -g 1218492533 -q 0.05 call-summits --nomodel --shift 37 --extsize 73`". Consensus peaksets between biological replicates were determined by intersecting the peaksets of each replicate by using the BEDTools intersect function and only keeping peaks present in both replicates. Similar to CUT&RUN data processing, featureCounts was used to generate peak count matrices for specific BAM files (when applicable) using the following arguments "`--fracOverlap 0.5 --minOverlap 5 -p`" and the edgeR package was used to first obtain TMM-normalized effective library sizes that were then used to normalize raw peak counts with the `cpm()` function. Peaks were annotated based on their closest gene using the R package ChIPSeeker and tornado and profile plots were generated using the DeepTools package. The bamCoverage function of DeepTools was used to generate RPGC-normalized BigWig files, for visualization of normalized read pile-up in a genome browser (IGV v2.13.0), using the following arguments "`--outFileFormat bigwig --binSize 5 --numberOfProcessors 10 --normalizeUsing RPGC --effectiveGenomeSize 1218492533 --extendReads`".

### Timecourse ATAC-seq data analysis

Publicly available time-course bulk ATAC-seq data from sorted NCCs isolated at different developmental stages were used to perform this analysis. The fastq files corresponding to biological duplicates for HH6, HH8, and HH10 stages were obtained and processed initially as described previously in the "ATAC-seq data analysis" section. The `wiggletools mean` function was used to generate bigwig files representing the average signal for each set of biological replicates corresponding to a specific stage. The consensus peaksets from the PanKla and H3K27ac CUT&RUNs were then used to generate profile plots displaying the cumulative ATAC-seq signal at each developmental stage as described previously. To quantify the increase in accessibility across the three developmental stages (at PanKla peaks) an aggregate peakset of all the ATAC-seq samples/replicates was generated and the `featureCounts` function was used to generate a peak count matrix (using the aggregate peakset a reference) with the following arguments "`--fracOverlap 0.5 --minOverlap 5 -p`". The edgeR package was then employed to obtain TMM-normalized effective library sizes that were used to normalize raw peak counts with the `cpm()` function. A subset of lactylated peaks was created from the aggregate peakset and their ATAC-seq signal (at the three different stages) was plotted.

### scATAC-seq data analysis

All data processing steps were performed using the Cell Ranger ATAC software[59]. Demultiplexing of raw files was done using `cellranger-atac mkfastq`, followed by filtering, alignment, counting, barcode counting, and cell calling via `cellranger-atac count`. Reads were aligned to the GRCg6a (GCA_000002315.5) chicken reference genome. Lastly, individual biological replicates were aggregated using the `cellranger-atac aggr` command. Downstream processing steps, including additional filtering of low-quality cells/outliers, normalization, dimensionality reduction, and clustering were performed using ArchR[60]. Cell datasets were additionally filtered by the total number of fragments in peaks per cell (>1000) and transcriptional start site score (score >3). Putative doublets were also removed using ArchR's doublet inference tool. Following TF-IDF normalization and dimensionality reduction, cells were visualized using a UMAP projection and clustered. LSI components displaying high correlation with sequencing depth were omitted from non-linear dimension reduction and clustering. Cells from different biological replicates were batch corrected using Harmony[61]. Upon clustering of data, peaks were re-called on a per cluster basis using MACS2. Clusters were empirically assigned using marker genes identified through motif and gene score enrichment across the UMAP clusters. Enrichment of lactylated peaks within the scATAC-seq UMAP was visualized using ArchR's addPeakAnnotations function to annotate scATAC-seq peaks overlapping lactylated regions. Next, ArchR's addDeviationsMatrix function was used to compute per-cell deviations of lactylated regions across the scATAC-seq dataset and the resulting deviation $z$-scores were projected onto the UMAP.

### DMSO vs. (R)-GNE-140 differential accessibility analysis

The R package DiffBind (version 2.14.0) was used to perform the differential accessibility analysis between DMSO and (R)-GNE-140 ATAC-seq samples. The consensus control peakset between DMSO replicates (generated using BEDTools as described in the "ATAC-seq data analysis" section) was first filtered to include only peaks that map to chromosomes (not contigs). This consensus peakset was then used to count reads from each sample/replicate. Binding affinity matrix scores were reported after performing a TMM normalization using full library size. The consensus peakset was filtered to remove peaks with low read counts (`filter=50` with `filterFun` applied to the scores of each interval). The comparison between samples was set up to account for the fact that the DMSO and (R)-GNE-140 samples from each set of biological replicates were paired (i.e., tissue came from the same set of embryos for each condition). This was done by specifying the `block=DBA_REPLICATE` argument when setting up contrasts using the `dba.contrast()` function. Differential accessibility statistical analysis was performed using the `DBA_EDGER` method. Assignment of differential peaks to their closest gene was done using the R package ChIPseeker.

### Control MO vs. SOX9 MO differential accessibility analysis

Differential accessibility analysis between GTBlue-Control MO and SOX9 MO ATAC-seq samples was performed using the R package DiffBind (version 2.14.0). Peak files corresponding to each sample/replicate were first filtered to include only peaks that map to chromosomes (not contigs). Reads from each sample/replicate were counted at a consensus peakset containing peaks that were present in at least 2 out of the 4 samples. This was done by specifying `minOverlap=2` in the `dba.count()` function. Binding affinity matrix scores were reported after performing a TMM normalization using full library size. The consensus peakset was filtered to remove peaks with low read counts (`filter=50` with `filterFun` applied to the scores of each interval). The comparison between samples was set up with the `dba.contrast()` function accounting for the fact that the samples were unpaired (i.e., Control MO and SOX9 MO transfected NFS came from different embryos). Differential accessibility statistical analysis was performed using the `DBA_EDGER` method. Assignment of differential peaks to their closest gene was done using the R package ChIPseeker.

### Additional statistical analyses

The R programing language was used to perform all statistical analyses. Independent and paired two-tailed $t$-tests were used to compare means between samples wherever applicable. Normality assumption of data points was checked using the Shapiro-Wilk test (`shapiro.test()` function in R) when sample sizes were appropriate, by inspecting quantile-quantile plots of the dataset, and by visually inspecting histograms of datapoints and/or model residuals (for linear models). Homogeneity of variances was checked using Levene's Test (`leveneTest()` function of `car` package in R) or by using the `var.test()` function in R. Whenever the variances between samples were not equal, a two-tailed Welch's $t$-test was used to compare the means. Non-normal data was log- ($\log_{10}$) or square-root-transformed (sqrt) whenever possible aiming to achieve a more normal distribution so as to perform parametric tests. Whenever data was not normally distributed, even after transformation, non-parametric tests such as the Kruskal–Wallis test (`kruskal.test()` function), Wilcoxon signed rank, or wilcoxon rank sum (`wilcox.test()` or `pairwise.wilcox.test()` functions) test were performed using R. When performing ad hoc pairwise Wilcoxon signed rank tests for more than two groups, $p$ values were corrected for multiple comparisons using the FDR approach (available through the `pairwise.wilcox.test()` function). Chi-square tests with Yates' continuity correction (`chisq.test()` function) were used to test for the independence between two categorical variables when dealing with count data. Simple linear and multiple regressions were run using the `lm()` function in R and results were summarized using the `summary()` function.

### Reporting summary

Further information on research design is available in the Nature Portfolio Reporting Summary linked to this article.

## Data availability

All datasets generated in this article have been deposited to the Gene Expression Omnibus: GSE228343. Source data are provided with this paper.

## Code availability

Scripts, code, and materials are available upon request.

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

## Acknowledgements

We thank P. Schweitzer and the Biotechnology Resource Center (BRC) Genomic Facility (RRID:SCR_021727) at the Cornell Institute of Biotechnology for help with NGS sequencing. This project was supported by the American Heart Association Pre-Doctoral Fellowship #915601 to F.M.

## Author contributions

Conceptualization: F.M. and M.S.C.; Methodology: F.M., M.R., and M.S.C.; Investigation: F.M. and M.R.; Visualization: F.M., M.R., and M.S.C.; Funding acquisition: M.S.C. and F.M.; Project administration: M.S.C.; Supervision: M.S.C.; Writing: F.M. and M.S.C.

## Competing interests

The authors declare no competing interests.
