## [Peer Review File · Nature Communications]

Histone lactylation couples cellular metabolism with developmental gene regulatory networksREVIEWER COMMENTS

Reviewer #1 (Remarks to the Author):

In this article Merkuri et al. study the role of histone lactylation in the control of gene expression, in particular in migratory cells of the neural crest, but also in cells of the presomitic mesoderm. Some functional approaches, such as inhibition of LDHA and LDHB, demonstrate the involvement of the lactylation process in the control of gene expression of NCC GRN genes as well as in the function of neural crest migratory cells. Finally, the authors show that the genomic regions where histone lactylation is located are probably mostly active enhancers in which the TFBS of Sox and TEAD are enriched.

I find this study very interesting, since it directly links metabolism with the control of gene expression, and more importantly with cell function, particularly in an extremely important tissue in vertebrates such as the NCCs. The different approaches used by the authors, combining different epigenomic techniques (Cut&Run, ATAC-seq, etc) with scATACseq, and with functional approaches in the chicken embryo are well suited for the questions they ask. The manuscript is correctly presented and easy to understand, and will be probably of interest for researchers from different fields.

I have several minor comments (below), but also one major comment (below), that I think should be corrected or modified before the manuscript can be published.

For all these reasons, I will only accept the manuscript after the corrections have been made.

Major comment:

The authors state that Sox9 and YAP/TEAD factors are necessary and sufficient for lactylation. In fact, they state that these transcription factors are the ones that drive lactylation specifically in NCCs. In my view, the authors do not directly prove these assertions. The approach they take is redundant, since they first look for enriched TFBS in the lactylated genomic zones, find Sox and YAP sites, and then use these two factors to model PanKla levels. The model shows that the relationship between PanKla and YAP1/Sox9 explains the signal variation. Similarly, using ATAC-seq in embryos in which Sox9 has been downregulated, they show that the ATAC peaks that disappear in these embryos have high levels of lactylation, which is again redundant, since high lactylation peaks possess significant Sox9 binding sites. And the experiment that I consider that could have contributed more to this question, which is the overexpression of Sox9 (and TEA1-VPR), is presented as a general normalized increase in lactylation, when they could have (in addition to what was shown in general), focused on one or several loci and shown that indeed, the binding of Sox9 precedes and drives the site lactylation. For these reasons, I suggest the authors to moderate their conclusions regarding the role of these two transcription factors in the lactylation process.

Minor comments

L28- "...and neural crest cells migration was impaired..."

L28- "The deposition of lactyl groups on hisytone at neural..."

Fig1E: Some non NCC show higher lactylation. Do the authors have an idea of which are these cells?

Fig1D: I think it should be better if colours of labelled vs unlabelled cells in this figure can be changed.

The colours used do not present the data at their best, particularly in printed versions of the paper.

L171- "These results show that at least part of the cis-regulatory"

L213 and L215- The authors present the results as "...increase progressively..." or "...is lower in magnitude..." without quantification

L230 and fig 3F- the authors say that inhibition of lactate production led to widespread changes in chromatin accessibility, but I cannot see these changes in the figure 3F (the three profiles are extremely similar)

L375- "...histone acetylations have..."

Reviewer #2 (Remarks to the Author):

The manuscript by Merkuri explores the role of histone lactylation on chromatin accessibility and gene

expression focussing on cranial neural crest cells in chick embryos. The work builds on a previous report from the group that YAP/TEAD acts downstream of glycolysis in NCCs. Here they cut&run to examine PanK1a (Lysine lactylated) and find that many neural crest genes are lactylated after increase of glycolysis during development, at Hamburger Hamilton stage 9/10. They find that PanK1a peaks correlate with ATAC peaks and that many of the lactylated gene loci are part of the neural crest gene network. In many cases histone lactylation correlated with active enhancers, three examples are shown using a GFP enhancer reporter plasmid. The lactylation mark is highly correlated with NCC genes at the EMT transition, and dependent on the activity of two enzymes responsible for synthesizing lactate from pyruvate. Pharmacological inhibition of LDHA and LDHB in neural crest explants reduced accessibility at lactylated genomic loci, and morpholino mediated LDHA/LDHB knock-down inhibited NCC migration. Interestingly, the lactylation mark is also present in presegmented mesoderm, another tissue which displays a glycolytic switch in development, and comparisons revealed some common and some unique PanK1a peaks. Analysis of enriched GO-terms lead the authors to propose that a glycolysis-lactylation axis is important for EMT and cell motility. Because they identify a strong enrichment for SOX transcription factor binding motifs in they performed CUT&RUN for SOX9, which showed correlations with their previous CUT&RUN peaks for active YAP1 in NCCs and with PanK1a peaks. MO inhibition of SOX9 led to widespread changes in chromatin accessibility, while overexpression of SOX9 plus TEA1-VPR enhanced lactylation. The authors may consider overexpression of SOX9 or TEA1-VPR individually to examine the effects on H3K18La and PanK1a. Having said this, overall the work is highly original and well conducted. It will be of broad interest not only to the fields of developmental and stem cell biology, but also to those working on the glycolytic switch in cancers.

Please ensure that all analytical tools used are referenced, e.g. chromVAR has no reference, as may some other tools.

Figure 1D, the authors comment on PanK1a staining. It appears notochord cells also display high levels of lactylation.

Figure 1J and 2C are partly redundant both show the same locus, the authors could consider showing another locus as an example.

Figure 3E-G, it is not clear in text description whether this is single cell ATAC or bulk ATAC from the NC explants.

Figure 4E provide a reference for chromVAR

Minor/typos:

Throughout you use 'lactylation deposition', is it either 'lactylation' or 'lactyl deposition'? see also line 324 "deposition of lactyl" rather than 'deposition of lactylation'

Line 55 display show – delete one

Line 71 as a model

Line 79 to survey histone

Line 96 glycolysis

Line 117 at HH9 during (transition?) to EMT

Reviewer #3 (Remarks to the Author):

Merkuri et al. present a detailed study of histone lactylation during development of neural crest and presomitic mesoderm. They show that this lactylation associates primarily with active enhancer regions, in a regulated and cell-type specific manner. Functional studies suggest that neural crest cell migration is significantly reduced in the absence of lactylation, and enhanced by increased availability of lactate. Knockdown experiments indicate that SOX9 and YAP/TEAD are important factors of neural-

crest specific histone lactylation. Overall, this study presents a compelling mechanistic link between cellular metabolism and regulation of important gene regulatory networks.

I find myself in the rare position of having little to say about this manuscript other than “well done.” The experiments are well-controlled and well-executed, and copious detailed data are provided in the supplemental files. The figures are clear and the authors are to be especially commended for their inclusion of cartoons outlining the main experiment for each figure as the first panel in each, making the flow of the experiments easy to follow. The only minor criticism I have is that the ordering of the panels in several figures (e.g., Fig 1, Fig 5) is confusing. I understand why this was done in terms of layout purposes, but suggest that maybe this is one of the unusual instances where ordering the panels sequentially and just referring to them out-of-order in the text might make more sense—I had to hunt around for the correct panel more than I would have liked.

Overall however I found this to be a clear and convincing study of high overall significance, and suitable for publication.

Response to Reviewer's Comments:

Reviewer #1:

In this article Merkuri et al. study the role of histone lactylation in the control of gene expression, in particular in migratory cells of the neural crest, but also in cells of the presomitic mesoderm. Some functional approaches, such as inhibition of LDHA and LDHB, demonstrate the involvement of the lactylation process in the control of gene expression of NCC GRN genes as well as in the function of neural crest migratory cells. Finally, the authors show that the genomic regions where histone lactylation is located are probably mostly active enhancers in which the TFBS of Sox and TEAD are enriched.

I find this study very interesting, since it directly links metabolism with the control of gene expression, and more importantly with cell function, particularly in an extremely important tissue in vertebrates such as the NCCs. The different approaches used by the authors, combining different epigenomic techniques (Cut&Run, ATAC-seq, etc) with scATACseq, and with functional approaches in the chicken embryo are well suited for the questions they ask. The manuscript is correctly presented and easy to understand, and will be probably of interest for researchers from different fields.

I have several minor comments (below), but also one major comment (below), that I think should be corrected or modified before the manuscript can be published.

For all these reasons, I will only accept the manuscript after the corrections have been made.

We thank the reviewer for highlighting the broad impact of our study.

Major comment:

The authors state that Sox9 and YAP/TEAD factors are necessary and sufficient for lactylation. In fact, they state that these transcription factors are the ones that drive lactylation specifically in NCCs. In my view, the authors do not directly prove these assertions. The approach they take is redundant, since they first look for enriched TFBS in the lactylated genomic zones, find Sox and YAP sites, and then use these two factors to model PanK1a levels. The model shows that the relationship between PanK1a and YAP1/Sox9 explains the signal variation. Similarly, using ATAC-seq in embryos in which Sox9 has been downregulated, they show that the ATAC peaks that disappear in these embryos have high levels of lactylation, which is again redundant, since high lactylation peaks possess significant Sox9 binding sites. And the experiment that I consider that could have

contributed more to this question, which is the overexpression of Sox9 (and TEA1-VPR), is presented as a general normalized increase in lactylation, when they could have (in addition to what was shown in general), focused on one or several loci and shown that indeed, the binding of Sox9 precedes and drives the site lactylation. For these reasons, I suggest the authors to moderate their conclusions regarding the role of these two transcription factors in the lactylation process.

We agree with the reviewer about the limitations of the study and have moderated our conclusions regarding the role of SOX9 and YAP/TEAD in the deposition of the lactylation mark. We now state that these transcriptional regulators facilitate the deposition of lactyl-CoA, instead of affirming that they are the cell-type specific drivers of lactylation. These changes appear in lines 376-377 and 394-396 (in the Results section) as well as in lines 429-430 and 448-449 and 462-464 (in the Discussion section). To further address the concerns of the reviewer, we also expanded the analysis of this experiment to show changes in lactylation levels at *loci* of specific neural crest genes.

Minor comments

L28- "...and neural crest cells migration was impaired..."

The typo was fixed.

L28- "The deposition of lactyl groups on hisytone at neural..."

The typo was fixed.

Fig1E: Some non NCC show higher lactylation. Do the authors have an idea of which are these cells?

We now comment on the non-NCCs with high lactylation levels (which include cells of the notochord and some cells in the neural tube) in the Results section of the manuscript (lines 109-110).

Fig1D: I think it should be better if colours of labelled vs unlabelled cells in this figure can be changed. The colours used do not present the data at their best, particularly in printed versions of the paper.

We agree with the reviewer and have now changed the pseudocolor that represents the intensity of PanK1a fluorescence to a blue-green scheme that is much clearer in printed versions of the manuscript.

L171- "These results show that at least part of the cis-regulatory"

Typo was fixed.

L213 and L215- The authors present the results as "...increase progressively..." or "...is lower in magnitude..." without quantification

We thank the reviewer for this comment. We have now included a supplemental figure (Supplemental Figure 3I – 3J) where we show the TMM normalized ATAC-seq signal at lactylated peaks at the three stages of NCC development. We use this data to perform a Kruskal-Wallis test followed by an *ad hoc* Wilcoxon signed-rank test (given the non-normal distribution of the datasets) to show that the increase in ATAC-seq signal at lactylated peaks is indeed statistically significant.

L230 and fig 3F- the authors say that inhibition of lactate production led to widespread changes in chromatin accessibility, but I cannot see these changes in the figure 3F (the three profiles are extremely similar)

We have now clarified this statement in the manuscript. The differences that we are highlighting in Figure 3F refer to the height of the ATAC-seq peaks and not necessarily their presence/absence in the respective conditions.

L375- "...histone acetylations have..."

Here we refer to histone acylations, the chemical family that acetylation, lactylation, and crotonylation belong to.

Reviewer #2:

The manuscript by Merkuri explores the role of histone lactylation on chromatin accessibility and gene expression focussing on cranial neural crest cells in chick embryos. The work builds on a previous report from the group that YAP/TEAD acts downstream of glycolysis in NCCs. Here they cut&run to examine PanKla (Lysine lactylated) and find that many neural crest genes are lactylated after increase of glycolysis during development, at Hamburger Hamilton stage 9/10. They find that PanKla peaks correlate with ATAC peaks and that many of the lactylated gene loci are part of the neural crest gene network. In many cases histone lactylation correlated with active enhancers, three examples are shown using a GFP enhancer reporter plasmid. The lactylation mark is highly correlated with NCC genes at the EMT transition, and dependent on the activity of two enzymes responsible for synthesizing lactate from pyruvate. Pharmacological inhibition of LDHA and LDHB in neural crest explants reduced accessibility at lactylated genomic loci, and morpholino mediated LDHA/LDHB knock-down inhibited NCC migration.

Interestingly, the lactylation mark is also present in presegmented mesoderm, another tissue which displays a glycolytic switch in development, and comparisons revealed some common and some unique PanKla peaks. Analysis of enriched GO-terms lead the authors to propose that a glycolysis-lactylation axis is important for EMT and cell motility. Because they identify a strong enrichment for SOX transcription factor binding motifs in they performed CUT&RUN for SOX9, which showed correlations with their previous

CUT&RUN peaks for active YAP1 in NCCs and with PanKla peaks. MO inhibition of SOX9 led to widespread changes in chromatin accessibility, while overexpression of SOX9 plus TEA1-VPR enhanced lactylation. The authors may consider overexpression of SOX9 or TEA1-VPR individually to examine the effects on H3K18La and PanKla.

Having said this, overall the work is highly original and well conducted. It will be of broad interest not only to the fields of developmental and stem cell biology, but also to those working on the glycolytic switch in cancers.

We thank the reviewer for underscoring the broad impact and strengths of our study.

Please ensure that all analytical tools used are referenced, e.g. chromVAR has no reference, as may some other tools.

We have included the references for all pipelines for data analysis used in the study, including chromVAR. All the tools that we have used are also referenced in the Methods sections. We thank the reviewer for pointing out this oversight.

Figure 1D, the authors comment on PanKla staining. It appears notochord cells also display high levels of lactylation.

We now include this information in the Results section (please see comment from Reviewer 1).

Figure 1J and 2C are partly redundant both show the same locus, the authors could consider showing another locus as an example.

We show the SNAI2 locus in the two figures because we wanted to compare the deposition of PanKla and H3k27ac at the same genomic location. We have however included additional *loci* in Supplemental Figure 1.

Figure 3E-G, it is not clear in text description whether this is single cell ATAC or bulk ATAC from the NC explants.

We thank the reviewer for pointing this out. We have clarified this in the text (line 232) as well as in the figure diagrams (Figure 3D, 3E) that we are using previously published datasets of bulk ATAC-seq performed from neural crest cells isolated with FACS from embryos at different developmental stages (Hovland et al., Dev Cell, 2022).

Figure 4E provide a reference for chromVAR

Reference has been added to the main text.

Minor/typos:

Throughout you use 'lactylation deposition', is it either 'lactylation' or 'lactyl

deposition’? see also line 324 “deposition of lactyl” rather than ‘deposition of lactylation’

We thank the reviewer for pointing this out. We have adjusted the text as necessary.

Line 55 display show – delete one

Line 71 as a model

Line 79 to survey histone

Line 96 glycolysis

Line 117 at HH9 during (transition?) to EMT

These mistakes have been corrected.

Reviewer #3:

Merkuri et al. present a detailed study of histone lactylation during development of neural crest and presomitic mesoderm. They show that this lactylation associates primarily with active enhancer regions, in a regulated and cell-type specific manner. Functional studies suggest that neural crest cell migration is significantly reduced in the absence of lactylation, and enhanced by increased availability of lactate. Knockdown experiments indicate that SOX9 and YAP/TEAD are important factors of neural-crest specific histone lactylation. Overall, this study presents a compelling mechanistic link between cellular metabolism and regulation of important gene regulatory networks.

I find myself in the rare position of having little to say about this manuscript other than “well done.” The experiments are well-controlled and well-executed, and copious detailed data are provided in the supplemental files. The figures are clear and the authors are to be especially commended for their inclusion of cartoons outlining the main experiment for each figure as the first panel in each, making the flow of the experiments easy to follow.

We thank the reviewer for their positive comments.

The only minor criticism I have is that the ordering of the panels in several figures (e.g., Fig 1, Fig 5) is confusing. I understand why this was done in terms of layout purposes, but suggest that maybe this is one of the unusual instances where ordering the panels sequentially and just referring to them out-of-order in the text might make more sense—I had to hunt around for the correct panel more than I would have liked.

Overall however I found this to be a clear and convincing study of high overall significance, and suitable for publication.

We agree with the reviewer that in some instances it might be a bit hard to locate the panel. We have rearranged Figure 5 per the reviewer's suggestions. In Figure 1 we opted to keep the current layout as we wanted the section of the embryonic head to be adjacent to the diagram schematizing NCC development.

REVIEWERS' COMMENTS

Reviewer #1 (Remarks to the Author):

After rereading the manuscript and the authors' responses, all my criticisms have been answered, in particular my major criticism. In this case the authors have moderated their conclusions in different parts of the manuscript and have included new analyses on specific neural crest gene loci, which go in the direction of their conclusions. For these reasons, I consider that the paper can be published in its present form in Nat Comm.

Reviewer #2 (Remarks to the Author):

This is a very nice paper with broad relevance to many fields not just neural crest development. The study is well conducted and presented and all my comments have been addressed.